# Truncated Linear Regression in High Dimensions

**Constantinos Daskalakis**
MIT
costis@mit.edu

**Dhruv Rohatgi**
MIT
drohatgi@mit.edu

**Manolis Zampetakis**
MIT
mzampet@mit.edu

## Abstract

As in standard linear regression, in *truncated* linear regression, we are given access to observations $(A_i, y_i)_i$ whose dependent variable equals $y_i = A_i^{\mathrm{T}} \cdot x^* + \eta_i$, where $x^*$ is some fixed unknown vector of interest and $\eta_i$ is independent noise; except we are only given an observation if its dependent variable $y_i$ lies in some "truncation set" $S \subset \mathbb{R}$. The goal is to recover $x^*$ under some favorable conditions on the $A_i$'s and the noise distribution. We prove that there exists a computationally and statistically efficient method for recovering $k$-sparse $n$-dimensional vectors $x^*$ from $m$ truncated samples, which attains an optimal $\ell_2$ reconstruction error of $O(\sqrt{(k \log n)/m})$. As a corollary, our guarantees imply a computationally efficient and information-theoretically optimal algorithm for compressed sensing with truncation, which may arise from measurement saturation effects. Our result follows from a statistical and computational analysis of the Stochastic Gradient Descent (SGD) algorithm for solving a natural adaptation of the LASSO optimization problem that accommodates truncation. This generalizes the works of both: (1) Daskalakis et al. [9], where no regularization is needed due to the low-dimensionality of the data, and (2) Wainright [27], where the objective function is simple due to the absence of truncation. In order to deal with both truncation and high-dimensionality at the same time, we develop new techniques that not only generalize the existing ones but we believe are of independent interest.

## 1 Introduction

In the vanilla linear regression setting, we are given $m \geq n$ observations of the form $(A_i, y_i)$, where $A_i \in \mathbb{R}^n$, $y_i = A_i^{\mathrm{T}} x^* + \eta_i$, $x^*$ is some unknown coefficient vector that we wish to recover, and $\eta_i$ is independent and identically distributed across different observations $i$ random noise. Under favorable conditions about the $A_i$'s and the distribution of the noise, it is well-known that $x^*$ can be recovered to within $\ell_2$-reconstruction error $O(\sqrt{n/m})$.

The classical model and its associated guarantees might, however, be inadequate to address many situations which frequently arise in both theory and practice. We focus on two common and widely studied deviations from the standard model. First, it is often the case that $m \ll n$, i.e. the number of observations is much smaller than the dimension of the unknown vector $x^*$. In this "underdetermined" regime, it is fairly clear that it is impossible to expect a non-trivial reconstruction of the underlying $x^*$, since there are infinitely many $x \in \mathbb{R}^n$ such that $A_i^{\mathrm{T}} x = A_i^{\mathrm{T}} x^*$ for all $i = 1, \ldots, m$. To sidestep this impossibility, we must exploit additional structural properties that we might know $x^*$ satisfies. One such property might be *sparsity*, i.e. that $x^*$ has $k \ll n$ non-zero coordinates. Linear regression under sparsity assumptions has been widely studied, motivated by applications such as model selection and compressed sensing; see e.g. the celebrated works of [24, 6, 11, 27] on this topic. It is known, in particular, that a $k$-sparse $x^*$ can be recovered to within $\ell_2$ error $O(\sqrt{k \log n/m})$, when the $A_i$'s are drawn from the standard multivariate Normal, or satisfy other favorable conditions [27]. The recovery algorithm solves a least squares optimization problem with $\ell_1$ regularization, i.e. what is called LASSO optimization in Statistics, in order to reward sparsity.

Another common deviation from the standard model is the presence of *truncation*. Truncation occurs when the sample $(A_i, y_i)$ is not observed whenever $y_i$ falls outside of a subset $S \subseteq \mathbb{R}$. Truncation arises quite often in practice as a result of saturation of measurement devices, bad data collection practices, incorrect experimental design, and legal or privacy constraints which might preclude the use of some of the data. Truncation is known to affect linear regression in counter-intuitive ways, as illustrated in Fig. 1, where the linear fits obtained via least squares regression before and after

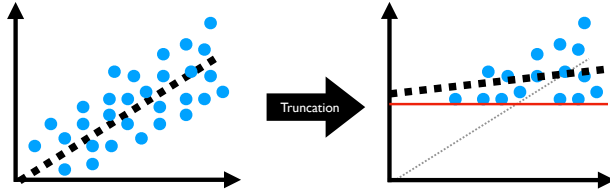

Figure 1: Truncation in one-dimensional linear regression, along with the linear fit obtained via least squares regression before and after truncation.

truncation of the data based on the value of the response variable are also shown. More broadly, it is well-understood that naive statistical inference using truncated data commonly leads to bias. Accordingly, a long line of research in Statistics and Econometrics has strived to develop regression methods that are robust to truncation [25, 1, 15, 19, 16, 5, 14]. This line of work falls into the broader field of Truncated Statistics [23, 8, 2], which finds its roots in the early works of [3], [13], [20, 21], and [12]. Despite this voluminous work, computationally and statistically efficient methods for truncated linear regression have only recently been obtained in [9], where it was shown that, under favorable assumptions about the $A_i$'s, the truncation set $S$, and assuming the $\eta_i$'s are drawn from a Gaussian, the negative log likelihood of the truncated sample can be optimized efficiently, and approximately recovers the true parameter vector with an $\ell_2$ reconstruction error $O\left(\sqrt{\frac{n \log m}{m}}\right)$.

**Our contribution.** In this work, we solve the general problem addressing both of the afore-described challenges together. Namely, we provide efficient algorithms for the high-dimensional ($m \ll n$) truncated linear regression problem. This problem is very common, including in compressed sensing applications with measurement saturation, as studied e.g. in [10, 17].

Under standard conditions on the design matrix and the noise distribution (namely that the $A_i$'s and $\eta_i$'s are sampled from independent Gaussian distributions before truncation), and under mild assumptions on the truncation set $S$ (roughly, that it permits a constant fraction of the samples $y_i = A_i^\mathrm{T} x^* + \eta_i$ to survive truncation), we show that the SGD algorithm on the *truncated LASSO* optimization program, our proposed adaptation of the standard LASSO optimization to accommodate truncation, is a computationally and statistically efficient method for recovering $x^*$, attaining an optimal $\ell_2$ reconstruction error of $O(\sqrt{(k \log n)/m})$, where $k$ is the sparsity of $x^*$.

We formally state the model and assumptions in Section 2, and our result in Section 3.

## 1.1 Overview of proofs and techniques

The problem that we solve in this paper encompasses the two difficulties of the problems considered in: (1) Wainwright [27], which tackles the problem of high-dimensional sparse linear regression with Gaussian noise, and (2) Daskalakis et al. [9], which tackles the problem of truncated linear regression. The tools developed in those papers do not suffice to solve our problem, since each difficulty interferes with the other. Hence, we introduce new ideas and develop new interesting tools that allow us to bridge the gap between [27] and [9]. We begin our overview in this section with a brief description of the approaches of [27, 9] and subsequently outline the additional challenges that arise in our setting, and how we address them.

Wainwright [27] uses as an estimator the solution of the regularized least squares program, also called *LASSO*, to handle the high-dimensionality of the data. Wainwright then uses a primal-dual witness method to bound the number of samples that are needed in order for the solution of the LASSO program to be close to the true coefficient vector $x^*$. The computational task is not discussed

in detail in Wainwright [27], since the objective function of the LASSO program is very simple and standard convex optimization tools can be used.

Daskalakis et al. [9] use as their estimator the solution to the log-likelihood maximization problem. In contrast to [27], their convex optimization problem takes a very complicated form due to the presence of truncation, which introduces an intractable log-partition function term in the log-likelihood. The main idea of Daskalakis et al. [9] to overcome this difficulty is identifying a convex set $D$ such that: (1) it contains the true coefficient vector $x^*$, (2) their objective function is strongly convex inside $D$, (3) inside $D$ there exists an efficient rejection sampling algorithm to compute unbiased estimates of the gradients of their objective function, (4) the norm of the stochastic gradients inside $D$ is bounded, and (5) projecting onto $D$ is efficient. These five properties are essentially what they need to prove that the SGD with projection set $D$ converges quickly to a good estimate of $x^*$.

Our reconstruction algorithm is inspired by both [27] and [9]. We formulate our optimization program as the $\ell_1$-regularized version of the negative log-likelihood function in the truncated setting, which we call *truncated LASSO*. In particular, our objective contains an intractable log-partition function term. Our proof then consists of two parts. First, we show statistical recovery, i.e. we upper bound the number of samples that are needed for the solution of the truncated LASSO program to be close to the true coefficient vector $x^*$. Second, we show that this optimization problem can be solved efficiently. The cornerstones of our proof are the two seminal approaches that we mentioned: the Primal-Dual Witness method for statistical recovery in high dimensions, and the Projected SGD method for efficient maximum likelihood estimation in the presence of truncation. Unfortunately, these two techniques are not a priori compatible to stitch together.

Roughly speaking, the technique of [9] relies heavily on the very carefully chosen *projection set* $D$ in which the SGD is restricted, as we explained above. This projection set cannot be used in high dimensions because it effectively requires knowing the low-dimension subspace in which the true solution lies. The projection set was the key to the aforementioned nice properties: strong convexity, efficient gradient estimation, and bounded gradient. In its absence, we need to deal with each of these issues individually. The primal-dual witness method of [27] cannot also be applied directly in our setting. In our case the gradient of the truncated LASSO does not have a nice closed form and hence finding the correct way to construct the primal-dual witness requires a more delicate argument. Our proof manages to overcome all these issues. In a nutshell the architecture of our full proof is the following.

1. **Optimality on the low dimensional subspace.** The first thing that we need to prove is that the optimum of the truncated LASSO program when restricted to the low dimensional subspace defined by the non-zero coordinates of $x^*$ is close to the true solution. This step of the proof was unnecessary in [9] due to the lack of regularization in their objective, and was trivial in [27] due to the simple loss function, i.e. the regularized least square.

2. **Optimality on the high dimensional space.** We prove that the optimum of the truncated LASSO program in the low dimensional subspace is also optimal for the whole space. This step is done using the primal-dual witness method as in [27]. However, in our case the expression of the gradient is much more complicated due to the very convoluted objective function. Hence, we find a more general way to prove this step that does not rely on the exact expression of the gradient.

These two steps of the proof suffice to upper bound the number of samples that we need to recover the coefficient vector $x^*$ via the truncated LASSO program. Next, we provide a computationally efficient method to solve the truncated LASSO program.

3. **Initialization of SGD.** The first step of our algorithm to solve truncated LASSO is finding a good initial point for the SGD. This was unnecessary in [27] due to the simple objective and in [9] due to the existence of the projection set $D$ (where efficient projection onto $D$ immediately gave an initial point). We propose the simple answer of *bootstrapping*: start with the solution of the $\ell_1$-regularized ordinary least squares program. This is a biased estimate, but we show it's good enough for initialization.

4. **Projection of SGD.** Next, we need to choose a projection set to make sure that Projected-SGD (PSGD) converges. The projection set chosen in [9] is not helpful in our case unless we a priori know the set of non-zero coordinates of $x^*$. Hence, we define a different,

simpler set which admits efficient projection algorithms. As a necessary side-effect, in contrast to [9], our set cannot guarantee many of the important properties that we need to prove fast convergence of SGD.

5. **Lack of strong convexity and gradient estimation**. Our different projection set cannot guarantee the strong convexity and efficient gradient estimation enjoyed in [9]. There are two problems here:

First, we know that PSGD converges to a point with small loss, but why must the point be near the optimum? Since strong convexity fails in high dimensions, it is not clear. We provide a workaround to resolve this issue that can be applied to other regularized programs with stochastic access to the gradient function.

Second, computing unbiased estimates of the gradient is now difficult. The prior work employed rejection sampling, but in our setting this may take exponential time. For this reason we provide a more explicit method for estimating the gradient much faster, whenever the truncation set is reasonably well-behaved.

An important tool that we leverage repeatedly in our analysis and we have not mentioned above is a strong isometry property for our measurement matrix, which has truncated Gaussian rows. Similar properties have been explored in the compressed sensing literature for matrices with i.i.d. Gaussian and sub-Gaussian entries [26].

We refer to Section 5 for a more detailed overview of the proofs of our main results.

**Organization.** Alphabetic sections (A, B, C, etc.) are located in the supplementary material.

## 2 High-dimensional truncated linear regression model

**Notation.** Let $Z \sim N(0,1)$ refer to a standard normal random variable. For $t \in \mathbb{R}$ and measurable $S \subseteq \mathbb{R}$, let $Z_t \sim N(t,1;S)$ refer to the truncated normal $(Z+t)|Z+t \in S$. Let $\mu_t = \mathbb{E}[Z_t]$. Let $\gamma_S(t) = \Pr[Z+t \in S]$. Additionally, for $a, x \in \mathbb{R}^n$ let $Z_{a,x}$ refer to $Z_{a^T x}$ (or $Z_{ax}$ if $a$ is a row vector), and let $\gamma_S(a,x)$ refer to $\gamma_S(a^T x)$. For a matrix $A \in \mathbb{R}^{m \times n}$, let $Z_{A,x} \in \mathbb{R}^m$ be the random vector with $(Z_{A,x})_j = Z_{A_j,x}$. For sets $I \subseteq [n]$ and $J \subseteq [m]$, let $A_{I,J}$ refer to the submatrix $[A_{i,j}]_{i \in I, j \in J}$. For $i \in [n]$ we treat the row $A_i$ as a row vector. In a slight abuse of notation, we will often write $A_U$ (or sometimes, $A_V$); this will *always* mean $A_{[m],U}$. By $A_U^T$ we mean $(A_{[m],U})^T$. For $x \in \mathbb{R}^n$, define $\mathrm{supp}(x)$ to be the set of indices $i \in [n]$ such that $x_i \neq 0$.

### 2.1 Model

Let $x^* \in \mathbb{R}^n$ be the unknown parameter vector which we are trying to recover. We assume that it is $k$-sparse; that is, $\mathrm{supp}(x^*)$ has cardinality at most $k$. Let $S \subseteq \mathbb{R}$ be a measurable subset of the real line. The main focus of this paper is the setting of *Gaussian noise*: we assume that we are given $m$ truncated samples $(A_i, y_i)$ generated by the following process:

1. Pick $A_i \in \mathbb{R}^n$ according to the standard normal distribution $N(0,1)^n$.
2. Sample $\eta_i \sim N(0,1)$ and compute $y_i$ as

$$y_i = A_i x^* + \eta_i. \tag{1}$$

3. If $y_i \in S$, then return sample $(A_i, y_i)$. Otherwise restart the process from step 1.

We also briefly discuss the setting of *arbitrary noise*, in which $\eta_i$ may be arbitrary and we are interested in approximations to $x^*$ which have guarantees bounded in terms of $\|\eta\|_2$.

Together, $m$ samples define a pair $(A, y)$ where $A \in \mathbb{R}^{m \times n}$ and $y \in \mathbb{R}^m$. We make the following assumptions about set $S$.

**Assumption I** (Constant Survival Probability)**.** Taking expectation over vectors $a \sim N(0,1)^n$, we have $\mathbb{E}\gamma_S(a,x^*) \geq \alpha$ for a constant $\alpha > 0$.

**Assumption II** (Efficient Sampling)**.** There is an $T(\gamma_S(t))$-time algorithm which takes input $t \in \mathbb{R}$ and produces an unbiased sample $z \sim N(t,1;S)$.

We do not require that $T(\cdot)$ is a constant, but it will affect the efficiency of our algorithm. To be precise, our algorithm will make $\text{poly}(n)$ queries to the sampling algorithm. As we explain in Lemma K.4 in Section K, if the set $S$ is a union of $r$ intervals $\cup_{i=1}^{r}[a_i, b_i]$, then the Assumption II is satisfied with $T(\gamma_S(t)) = \text{poly}(\log(1/\gamma_S(t)), r)$. We express the theorems below with the assumption that $S$ is a union of $r$ intervals in which case the algorithms have polynomial running time, but all the statements below can be replaced with the more general Assumption II and the running time changes from $\text{poly}(n, r)$ to $\text{poly}(n, T(e^{m/\alpha}))$.

## 3 Statistically and computationally efficient recovery

In this section we formally state our main results for recovery of a sparse high-dimensional coefficient vector from truncated linear regression samples. In Section 3.1, we present our result under the standard assumption that the error distribution is Gaussian, whereas in Section 3.2 we present our results for the case of adversarial error.

### 3.1 Gaussian noise

In the setting of Gaussian noise (before truncation), we prove the following theorem.

**Theorem 3.1.** *Suppose that Assumption I holds, and that we have $m$ samples $(A_i, y_i)$ generated from Process (1), with $n \geq m \geq Mk \log n$ for a sufficiently large constant $M$. Then, there is an algorithm which outputs $\bar{x}$ satisfying $\|\bar{x} - x^*\|_2 \leq O(\sqrt{(k \log n)/m})$ with probability $1 - 1/n - O(k \exp(-m^{1/5}))$. Furthermore, if the survival set $S$ is a union of $r$ intervals the running time of our algorithm is $\text{poly}(n, r)$.*

From now on, we will use the term "with high probability" when the rate of decay is not of importance. This phrase will always mean "with failure probability no worse than $O(1/n) + k \exp(-\Omega(m^{1/5}))$.

Observe that even without the added difficulty of truncation (e.g. if $S = \mathbb{R}$), sparse linear regression requires $\Omega(k \log n)$ samples by known information-theoretic arguments [27]. Thus, our sample complexity is information-theoretically optimal.

In one sentence, the algorithm optimizes the $\ell_1$-regularized sample negative log-likelihood via projected SGD. The negative log-likelihood of $x \in \mathbb{R}^n$ for a single sample $(a, y)$ is

$$\text{nll}(x; a, y) = \frac{1}{2}(a^T x - y)^2 + \log \int_S e^{-(a^T x - z)^2/2} \, dz.$$

Given multiple samples $(A, y)$, we can then write $\text{nll}(x; A, y) = \frac{1}{m} \sum_{j=1}^{m} \text{nll}(x; A_j, y_j)$. We also define the regularized negative log-likelihood $f : \mathbb{R}^n \to \mathbb{R}$ by $f(x) = \text{nll}(x; A, y) + \lambda \|x\|_1$. We claim that optimizing the following program approximately recovers the true parameter vector $x^*$ with high probability, for sufficiently many samples and appropriate regularization $\lambda$:

$$\min_{x \in \mathbb{R}^n} \text{nll}(x; A, y) + \lambda \|x\|_1. \tag{2}$$

The first step is to show that any solution to Program (2) will be near the true solution $x^*$. To this end, we prove the following theorem, which already shows that $O(k \log n)$ samples are sufficient to solve the problem of *statistical* recovery of $x^*$:

**Proposition 3.2.** *Suppose that Assumption I holds. There are constants[1] $\kappa$, $d$, and $\sigma$ with the following property. Suppose that $m > \kappa \cdot k \log n$, and let $(A, y)$ be $m$ samples drawn from Process 1. Let $\hat{x}$ be any optimal solution to Program (2) with regularization constant $\lambda = \sigma\sqrt{(\log n)/m}$. Then $\|\hat{x} - x^*\|_2 \leq d\sqrt{(k \log n)/m}$ with high probability.*

Then it remains to show that Program (2) can be solved efficiently. Theorem 3.1 follows immediately from the concatenation of Propositions 3.2 and 3.3.

**Proposition 3.3.** *Suppose that Assumption I holds and let $(A, y)$ be $m$ samples drawn from Process 1 and $\hat{x}$ be any optimal solution to Program (2). There exists a constant $M$ such that if $m \geq Mk \log n$ then there is an algorithm which outputs $\bar{x} \in \mathbb{R}^n$ satisfying $\|\bar{x} - \hat{x}\|_2 \leq O(\sqrt{(k \log n)/m})$ with high probability. Furthermore, if the survival set $S$ is a union of $r$ intervals the running time of our algorithm is $\text{poly}(n, r)$.*

We present a more detailed description of the algorithm that we use in Section 4.

### 3.2 Adversarial noise

In the setting of arbitrary noise, optimizing negative log-likelihood no longer makes sense, and indeed our results from the setting of Gaussian noise no longer hold. However, we may apply results from compressed sensing which describe sufficient conditions on the measurement matrix for recovery to be possible in the face of arbitrary bounded error. We obtain the following theorem:

**Theorem 3.4.** *Suppose that Assumption I holds and let $\epsilon > 0$. There are constants $c$ and $M$ such that if $m \geq Mk \log n$, $\|Ax^* - y\|_2 \leq \epsilon$, and $\hat{x}$ minimizes $\|x\|_1$ in the region $\{x \in \mathbb{R}^n : \|Ax - y\|_2 \leq \epsilon\}$, then $\|\hat{x} - x^*\|_2 \leq c\epsilon/\sqrt{m}$.*

The proof is a corollary of our result that $A$ satisfies the Restricted Isometry Property from [6] with high probability even when we only observe truncated samples; see Corollary G.6 and the subsequent discussion in Section G.

The remainder of the paper is dedicated to the case where the noise is Gaussian before truncation.

## 4 The efficient estimation algorithm

Define $\mathscr{E}_r = \{x \in \mathbb{R}^n : \|Ax - y\|_2 \leq r\sqrt{m}\}$. To solve Program 2, our algorithm is Projected Stochastic Gradient Descent (PSGD) with projection set $\mathscr{E}_r$, for an appropriate constant $r$ (specified in Lemma 5.4). We pick an initial feasible point by computing

$$x^{(0)} = \underset{x \in \mathscr{E}_r}{\text{argmin}} \|x\|_1.$$

Subsequently, the algorithm performs $N$ updates, where $N = \text{poly}(n)$. Define a random update to $x^{(t)} \in \mathbb{R}^n$ as follows. Pick $j \in [m]$ uniformly at random. Sample $z^{(t)} \sim Z_{A_j, x^{(i)}}$. Then set

$$v^{(t)} := A_j(z^{(t)} - y_j) + \lambda \cdot \text{sign}(x^{(t)})$$

$$w^{(t)} := x^{(t)} - \sqrt{\frac{1}{nN}} v^{(t)}; \qquad x^{(t+1)} := \underset{x \in \mathscr{E}_r}{\text{argmin}} \left\| x - w^{(t)} \right\|_2.$$

Finally, the algorithm outputs $\bar{x} = \frac{1}{N} \sum_{t=0}^{N-1} x^{(t)}$.

See Section 5.2 for the motivation of this algorithm, and a proof sketch of correctness and efficiency. Section E contains a summary of the complete algorithm in pseudocode.

## 5 Overview of proofs and techniques

This section outlines our techniques. The first step, outlined in Section 5.1, is proving Proposition 3.2. The second step, outlined in Section 5.2, is proving Proposition 3.3, by showing that the algorithm described in Section 4 efficiently recovers an approximate solution to Program 2.

### 5.1 Statistical recovery

Our approach to proving Proposition 3.2 is the Primal-Dual Witness (PDW) method introduced in [27]. Namely, we are interested in showing that the solution of Program (2) is near $x^*$ with high probability. Let $U$ be the (unknown) support of the true parameter vector $x^*$. Define the following (hypothetical) program in which the solution space is restricted to vectors with support in $U$:

$$\underset{x \in \mathbb{R}^n : \text{supp}(x) \subseteq U}{\text{argmin}} \quad \text{nll}(x; A, y) + \lambda \|x\|_1 \tag{3}$$

In the untruncated setting, the PDW method is to apply the subgradient optimality condition to the solution $\breve{x}$ of this restricted program, which is by definition sparse. Proving that $\breve{x}$ satisfies subgradient optimality for the original program implies that the original program has a sparse solution $\breve{x}$, and under mild extra conditions $\breve{x}$ is the unique solution. Thus, the original program recovers the true basis. We use the PDW method for a slightly different purpose; we apply subgradient optimality to $\breve{x}$ to show that $\|\breve{x} - x^*\|_2$ is small, and then use this to prove that $\breve{x}$ solves the original program.

Truncation introduces its own challenges. While Program 2 is still convex [9], it is much more convoluted than ordinary least squares. In particular, the gradient and Hessian of the negative log-likelihood have the following form (see Section C for the proof).

**Lemma 5.1.** *For all $(A, y)$, the gradient of the negative log-likelihood is $\nabla \operatorname{nll}(x; A, y) = \frac{1}{m} \sum_{j=1}^{m} A_j^T (\mathbb{E} Z_{A_j,x} - y_j)$. The Hessian is $H(x; A, y) = \frac{1}{m} \sum_{j=1}^{m} A_j^T A_j \operatorname{Var}(Z_{A_j,x})$.*

We now state the two key facts which make the PDW method work. First, the solution $\breve{x}$ to the restricted program must have a zero subgradient in all directions in $\mathbb{R}^U$. Second, if this subgradient can be extended to all of $\mathbb{R}^n$, then $\breve{x}$ is optimal for the original program. Formally:

**Lemma 5.2.** *Fix any $(A, y)$. Let $\breve{x}$ be an optimal solution to Program 3.*

(a) *There is some $\breve{z}_U \in \mathbb{R}^U$ such that $\|\breve{z}_U\|_\infty \le 1$ and $-\lambda \breve{z}_U = \frac{1}{m} A_U^T (\mathbb{E} Z_{A,\breve{x}} - y)$.*

(b) *Extend $\breve{z}_U$ to $\mathbb{R}^n$ by defining $-\lambda \breve{z}_{U^c} = \frac{1}{m} A_{U^c}^T (\mathbb{E} Z_{A,\breve{x}} - y)$. If $\|\breve{z}_{U^c}\|_\infty < 1$, and $A_U^T A_U$ is invertible, then $\breve{x}$ is the unique optimal solution to Program 2.*

See Section C for the proof. The utility of this lemma is in reducing Proposition 3.2 to showing that with high probability over $(A, y)$, the following conditions both hold:

1. $\|\breve{x} - x^*\|$ is small (Theorem A.7).
2. $\|\breve{z}_{U^c}\|_\infty < 1$ (Theorem B.3) and $A_U^T A_U$ is invertible (corollary of Theorem G.1).

In Section A, we prove Condition (1) by dissecting the subgradient optimality condition. In Section B we then prove Condition (2) and complete the proof of Proposition 3.2.

## 5.2 Computational recovery

To prove Proposition 3.3, we need to show how to efficiently solve Program 2, i.e. minimize $f(x) = \operatorname{nll}(x; A, y) + \lambda \|x\|_1$. The gradient of $\operatorname{nll}(x; A, y)$ doesn't have a closed form, but it can be written cleanly as an expectation:

$$\nabla \operatorname{nll}(x; A, y) = \frac{1}{m} \sum_{j=1}^{m} A_j^T (\mathbb{E} Z_{A_j,x} - y_j).$$

Let us assume that $Z_{A_j,x}$ can be sampled efficiently (Assumption II). Then we may hope to optimize the convex function $f(x)$ by stochastic gradient descent. But problematically, in our high-dimensional setting $f$ is nowhere strongly convex. Convex optimization results which do not require strong convexity have several strings attached:

**Theorem 5.3.** *Let $f : \mathbb{R}^n \to \mathbb{R}$ be a convex function achieving its optimum at $\breve{x} \in \mathbb{R}^n$. Let $x^{(0)}, x^{(1)}, \ldots, x^{(N)}$ be a sequence of random vectors in $\mathbb{R}^n$. Suppose that $x^{(i+1)} = x^{(i)} - \eta v^{(i)}$ where $\mathbb{E}[v^{(i)}|x^{(i)}] \in \partial f(x^{(i)})$. Set $\bar{x} = \frac{1}{N} \sum_{i=1}^{N} x^{(i)}$. Then*

$$\mathbb{E}[f(\bar{x})] - f(\breve{x}) \le (\eta N)^{-1} \mathbb{E}\left[\left\|x^{(0)} - \breve{x}\right\|_2^2\right] + \eta N^{-1} \sum_{i=1}^{N} \mathbb{E}\left[\left\|v^{(i)}\right\|_2^2\right].$$

In particular, to use to this result to show that SGD converges to a good estimate of $\breve{x}$ in a polynomial number of iterations, we need to solve three technical problems:

1. We need to efficiently find an initial point $x^{(0)}$ with bounded distance from $\breve{x}$.

2. The gradient does not have bounded norm for arbitrary $x \in \mathbb{R}^n$. Thus, we need to pick a projection set in which the bound holds, and project to this set at each SGD iteration. If $\breve{x}$ lies in the projection set, then the guarantees of Theorem 5.3 still hold (see Theorem H.1).

3. Since $f$ is not strongly convex, we need some other way of converting the bound on $f(\bar{x}) - f(\check{x})$ into a bound on $\|\bar{x} - \check{x}\|_2$.

As defined in Section 4, our chosen projection set is $\mathscr{E}_r = \{x \in \mathbb{R}^n : \|Ax - y\|_2 \leq r\sqrt{m}\}$, for an appropriate constant $r > 0$. To pick an initial point in $\mathscr{E}_r$, we solve the program $x^{(0)} = \operatorname{argmin}_{x \in \mathscr{E}_r} \|x\|_1$. This estimate is biased due to the truncation, but the key point is that by classical results from compressed sensing, it has bounded distance from $x^*$ (and therefore from $\check{x}$).

The algorithm then consists of projected stochastic gradient descent with projection set $\mathscr{E}_r$. To bound the number of update steps required for the algorithm to converge to a good estimate of $\check{x}$, we need to solve several statistical problems (which are direct consequences of assumptions in Theorem 5.3).

**Properties of $\mathscr{E}_r$.** First, $\check{x}$ must be feasible and a bounded distance from the initial point (due to high-dimensionality, $\mathscr{E}_r$ is unbounded, so this is not immediate). The following lemmas formalize this; see Sections J.11 and J.12 for the respective proofs. Lemma 5.4 specifies the choice of $r$.

**Lemma 5.4.** *With high probability, $\check{x} \in \mathscr{E}_r$ for an appropriate constant $r > 0$.*

**Lemma 5.5.** *With high probability, $\|x^{(0)} - \check{x}\|_2 \leq O(1)$.*

Second, the SGD updates at points within $\mathscr{E}_r$ must be unbiased estimates of the gradient, with bounded square-norm in expectation. The following lemma shows that the updates $v^{(t)}$ defined in Section 4 satisfy this property. See Section J.13 for the proof.

**Lemma 5.6.** *With high probability over $A$, the following statement holds. Let $0 \leq t < T$. Then $\mathbb{E}[v^{(t)} | x^{(t)}] \in \partial f(x^{(t)})$, and $\mathbb{E}[\|v^{(t)}\|_2^2] \leq O(n)$.*

**Addressing the lack of strong convexity.** Third, we need to show that the algorithm converges in parameter space and not just in loss. That is, from the preceding bounds it follows that the algorithm produces an estimate $x$ for which $f(x) - f(\check{x})$ is small, but we want to show that $\|x - \check{x}\|_2$ is small as well. Note that $f$ is not strongly convex even in $\mathscr{E}_r$, due to the high dimension. So we need a more careful approach. In the subspace $\mathbb{R}^U$, $f$ is indeed strongly convex near $\check{x}$, as shown in the following lemma (proof in Section J.14):

**Lemma 5.7.** *There is a constant $\zeta$ such that with high probability over $A$, $f(x) - f(\check{x}) \geq \frac{\zeta}{2} \|x - \check{x}\|_2^2$ for all $x \in \mathbb{R}^n$ with $\operatorname{supp}(x) \subseteq U$ and $\|x - \check{x}\|_2 \leq 1$.*

But we need a bound for all $\mathbb{R}^n$, since the SGD iterates in general may be dense vectors. The idea is to prove a lower bound on $f(x) - f(\check{x})$ for $x$ near $\check{x}$, and then use convexity to scale the bound linearly in $\|x - \check{x}\|_2$. The above lemma provides a lower bound for $x$ near $\check{x}$ if $\operatorname{supp}(x) \subseteq U$, and we need to show that adding an $\mathbb{R}^{U^c}$-component to $x$ increases $f$ proportionally. This is precisely the content of Theorem B.4. Putting these pieces together we obtain the following lemma. See Section J.15 for the full proof.

**Lemma 5.8.** *There are constants $c_f > 0$ and $c_f'$ such that with high probability over $A$ the following holds. Let $x \in \mathbb{R}^n$. If $f(x) - f(\check{x}) \leq c_f (\log n)/m^3$, then*

$$\|x - \check{x}\|_2^2 \leq c_f' \frac{m}{\log n} (f(x) - f(\check{x})).$$

**Convergence of PSGD.** It now follows from the above lemmas and Theorem 5.3 that the PSGD algorithm, as outlined above and described in Section 4, converges to a good approximation of $\check{x}$ in a polynomial number of updates. The following theorem formalizes the guarantee. See Section J.16 for the proof.

**Theorem 5.9.** *With high probability over $(A, y)$ and over the execution of the algorithm, we get*

$$\|\bar{x} - \check{x}\|_2 \leq O(\sqrt{(k \log n)/m}).$$

**Efficient implementation.** Finally, in Section F we prove that initialization and each update step is efficient. Efficient gradient estimation in the projection set (i.e. sampling $Z_{A_j,x}$) does not follow from the prior work, since our projection set is by necessity laxer than that of the prior work [9]. So we replace their rejection sampling procedure with a novel approximate sampling procedure under mild assumptions about the truncation set. Together with the convergence bound claimed in Theorem 5.9, these prove Proposition 3.3.

**Proof of Proposition 3.3.** The correctness guarantee follows from Theorem 5.9. For the efficiency guarantee, note that the algorithm performs initialization and then $N = \text{poly}(n)$ update steps. By Section F, the initialization takes $\text{poly}(n)$ time, and each update step takes $\text{poly}(n) + T(e^{-\Theta(m)})$. This implies the desired bounds on overall time complexity. $\qquad\square$

## Broader impact

Our contribution is mathematical and computational in nature. We show that truncated linear regression is computationally and statistically efficiently solvable even in the sparse, high-dimensional setting. Our advances in statistical inference with truncation present opportunities in decreasing the bias of models that are trained on biased data. Biased data is a real and significant problem in applications of Machine Learning, as well as broader data science applications and scientific reasoning. There are many types of bias and their source and type might be known or unknown. In this paper, we focus on bias coming from systematic data truncation, which is an important type of bias as discussed in the introduction. In extending prior work on this topic to high dimensions we address another growing challenge—namely, the dimensionality of the data which must now be processed. As usual, in interpreting our methodological advancements and using our algorithms in practice, we must recognize that the theory is only as good as the model: we make several assumptions (e.g. Gaussian measurements and noise), and the practitioner must judge whether these assumptions are justified in their application.

## Acknowledgments

This research was supported by NSF Awards IIS-1741137, CCF-1617730 and CCF-1901292, by a Simons Investigator Award, by the DOE PhILMs project (No. DE-AC05-76RL01830), by the DARPA award HR00111990021, by the MIT Undergraduate Research Opportunities Program, and by a Google PhD Fellowship.

## Footnotes

[1]In the entirety of this paper, constants may depend on $\alpha$.

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
