[Supplementary Material]

# supplementary material for paper: Truncated Linear Regression in High Dimensions

**Constantinos Daskalakis**
MIT
costis@mit.edu

**Dhruv Rohatgi**
MIT
drohatgi@mit.edu

**Manolis Zampetakis**
MIT
mzampet@mit.edu

## A  Bounding solution of restricted program

In this section we prove that $\|\breve{x} - x^*\|_2$ is small with high probability, where $\breve{x}$ is a solution to Program 3. Specifically, we use regularization parameter $\lambda = \Theta(\sqrt{(\log n)/m})$, and prove that $\|\breve{x} - x^*\|_2 \leq O(\sqrt{(k \log n)/m})$.

The proof is motivated by the following rephrasal of part (a) of Lemma 5.2:

$$-\lambda \breve{z}_U = \frac{1}{m} A_U^T (\mathbb{E}[Z_{A,\breve{x}}] - \mathbb{E}[Z_{A,x^*}]) + \frac{1}{m} A_U^T (\mathbb{E}[Z_{A,x^*}] - y) \qquad (4)$$

where $\|\breve{z}_U\|_\infty \leq 1$. For intuition, consider the untruncated setting: then $\mathbb{E}[Z_t] = t$, so the equation is simply

$$-\lambda \breve{z}_U = \frac{1}{m} A_U^T A_U (\breve{x}_U - x_U^*) - \frac{1}{m} A_U^T w$$

where $w \sim N(0,1)^m$. Since $w$ is independent of $A_U^T$ and has norm $\Theta(m)$, each entry of $A_U^T w$ is Gaussian with variance $\Theta(m)$, so $\frac{1}{m} A_U^T w$ has norm $\Theta(\sqrt{k/m})$. Additionally, $\|\lambda \breve{z}_U\|_2 \leq \lambda \sqrt{k} = O(\sqrt{(k \log n)/m})$. Finally, $\frac{1}{m} A_U^T A_U$ is a $\Theta(1)$-isometry, so we get the desired bound on $\breve{x}_U - x_U^*$.

Returning to the truncated setting, one bound still holds, namely $\|\lambda \breve{z}_U\|_2 \leq \lambda \sqrt{k}$. The remainder of the above sketch breaks down for two reasons. First, $\mathbb{E}[Z_{A,x^*}] - y$ is no longer independent of $A$. Second, bounding $\frac{1}{m} A_U^T (\mathbb{E}[Z_{A,\breve{x}}] - \mathbb{E}[Z_{A,x^*}])$ no longer implies a bound on $\breve{x}_U - x_U^*$.

The first problem is not so hard to work around; we can still bound $A_U^T (\mathbb{E}[Z_{A,x^*}] - y)$ as follows; see Section J.1 for the proof.

**Lemma A.1.** *With high probability over $A$ and $y$, $\left\| A_U^T (\mathbb{E}[Z_{A,x^*}] - y) \right\|_2^2 \leq \alpha^{-1} km \log n$.*

So in equation 4, the last term is $O(\sqrt{(k \log n)/m})$ with high probability. The first term is always $O(\sqrt{(k \log n)/m})$, since $\|\breve{z}_U\|_2 \leq \sqrt{k}$. So we know that $\frac{1}{m} A_U^T (\mathbb{E}[Z_{A,\breve{x}}] - \mathbb{E}[Z_{A,x^*}])$ has small norm. Unfortunately this does not imply that $\mathbb{E}[Z_{A,\breve{x}}] - \mathbb{E}[Z_{A,x^*}]$ has small norm, but as motivation, assume that we have such a bound.

Since $A_U$ is a $\Theta(\sqrt{m})$-isometry, bounding $\breve{x} - x^*$ is equivalent to bounding $A\breve{x} - Ax^*$. To relate this quantity to $\mathbb{E}[Z_{A,\breve{x}}] - \mathbb{E}[Z_{A,x^*}]$, our approach is to lower bound the derivative of $\mu_t = \mathbb{E}[Z_t]$ with respect to $t$. The derivative turns out to have the following elegant form (proof in Section J.2):

**Lemma A.2.** *For any $t \in \mathbb{R}$, $\frac{d}{dt} \mu_t = \text{Var}(Z_t)$.*

Crucially, $\text{Var}(Z_t)$ is nonnegative, and relates to survival probability. By integrating a lower bound on the derivative, we get the following lower bound on $\mu_t - \mu_{t^*}$ in terms of $t - t^*$. The bound is linear for small $|t - t^*|$, but flattens out as $|t - t^*|$ grows. See Section J.3 for the proof.

**Lemma A.3.** *Let $t, t^* \in \mathbb{R}$. Then $\text{sign}(\mu_t - \mu_{t^*}) = \text{sign}(t - t^*)$. Additionally, for any constant $\beta > 0$ there is a constant $c = c(\beta) > 0$ such that if $\gamma_S(t^*) \geq \beta$, then $|\mu_t - \mu_{t^*}| \geq c \min(1, |t - t^*|)$.*

If we want to use this lemma to prove that $\|\mathbb{E}[Z_{A,\breve{x}}] - \mathbb{E}[Z_{A,x^*}]\|_2$ is at least a constant multiple of $\|A(\breve{x} - x^*)\|_2$, we face two obstacles: (1) $\gamma_S(A_j x^*)$ may not be large for all $j$, and (2) the lemma only gives linear scaling if $|A_j(\breve{x} - x^*)| = O(1)$: but this is essentially what we're trying to prove!

To deal with obstacle (1), we restrict to the rows $j \in [m]$ for which $\gamma_S(A_j x^*)$ is large. To deal with obstacle (2), we have a two-step proof. In the first step, we use the $\Omega(1)$-lower bound provided by Lemma A.3 to show that $\|A(\breve{x} - x^*)\|_2 = O(\sqrt{m})$ (so that $|A_j(\breve{x} - x^*)| = O(1)$ on average). In the second step, we use this to get linear scaling in Lemma A.3, and complete the proof, showing that $\|A(\breve{x} - x^*)\|_2 = O(\sqrt{k \log n})$.

Formally, define $I_{\text{good}}$ to be the set of indices $j \in [m]$ such that $\gamma_S(A_j x^*) \geq \alpha/2$ and $|A_j x^* - A_j \breve{x}|^2 \leq (6/(\alpha m)) \|Ax^* - A\breve{x}\|^2$. In the following lemmas we show that $I_{\text{good}}$ contains a constant fraction of the indices, so by the isometry properties we retain a constant fraction of $\|A(\breve{x} - x^*)\|_2$ when restricting to $I_{\text{good}}$. See Appendices J.4 and J.5 for the proofs of Lemmas A.4 and A.5 respectively.

**Lemma A.4.** *With high probability, $|I_{good}| \geq (\alpha/6)m$.*

**Lemma A.5.** *For some constant $\epsilon > 0$, we have that with high probability, $\|Ax^* - A\breve{x}\|^2_{I_{good}} \geq \epsilon \|Ax^* - A\breve{x}\|^2$.*

We now prove the weaker, first-step bound on $\|A(\breve{x} - x^*)\|_2$. But there is one glaring issue we must address: we made a simplifying assumption that $\|\mathbb{E}[Z_{A,\breve{x}}] - \mathbb{E}[Z_{A,x^*}]\|$ is small. All we actually know is that $\|A_U^T(\mathbb{E}[Z_{A,\breve{x}}] - \mathbb{E}[Z_{A,x^*}])\|_2$ is small. And $A_U^T$ has a nontrivial null space.

Here is a sketch of how we resolve this issue. Let $a = A(\breve{x} - x^*)$ and $b = \mu_{A\breve{x}} - \mu_{Ax^*}$; we want to show that if $\|a\|$ is large then $\|A_U^T b\|$ is large. Geometrically, $\|A_U^T b\|$ is approximately proportional to the distance from $b$ to the subspace $\text{Null}(A_U^T)$. Oversimplifying for clarity, we know that $|b_j| \geq c|a_j|$ for all $j$. This is by itself insufficient. The key observation is that we also know $\text{sign}(a_j) = \text{sign}(b_j)$ for all $j$. Thus, $b$ lies in a hyperoctant shifted to have corner $ca$. Since $ca$ lies in the row space of $A_U^T$, it's perpendicular to $\text{Null}(A_U^T)$, so the closest point to $\text{Null}(A_U^T)$ in the shifted hyperoctant should be $ca$.

Formalizing this geometric intuition yields the last piece of the proofs of the following theorems. See Section J.6 for the full proofs.

**Theorem A.6.** *There are positive constants $c'_{reg} = c'_{reg}(\alpha)$, $M' = M'(\alpha)$, and $C' = C'(\alpha)$ with the following property. Suppose that $\lambda \leq c'_{reg}/\sqrt{k}$ and $m \geq M'k \log n$. Then with high probability, $\|A_U x^* - A_U \breve{x}\|_2 \leq C'\sqrt{m}$.*

**Theorem A.7.** *There are positive constants $c''_{reg} = c''_{reg}(\alpha)$, $M'' = M''(\alpha)$, and $C'' = C''(\alpha)$ with the following property. Suppose that $\lambda \leq c''_{reg}/\sqrt{k}$ and $m \geq M''k \log n$. Then $\|x^* - \breve{x}\|_2 \leq C''(\lambda\sqrt{k} + \sqrt{(k \log n)/m})$ with high probability.*

# B   Proof of statistical recovery

Extend $\breve{z}$ to $\mathbb{R}^n$ by defining

$$\breve{z}_{U^c} = -\frac{1}{\lambda m} A_{U^c}^T (\mathbb{E}Z_{A,\breve{x}} - y).$$

We would like to show that $\|z_{U^c}\|_\infty < 1$. Since $A_{U^c}^T$ is independent of $\mathbb{E}[Z_{A,\breve{x}}] - y$, each entry of $A_{U^c}^T(\mathbb{E}[Z_{A,\breve{x}}] - y)$ is Gaussian with standard deviation $\|\mathbb{E}[Z_{A,\breve{x}}] - y\|_2$. It turns out that a bound of $O(\lambda\sqrt{km} + \sqrt{m})$ suffices. To get this bound, we decompose

$$\mathbb{E}[Z_{A,\breve{x}}] - y = A(\breve{x} - x^*) + \mathbb{E}R_{A,\breve{x}} - (y - Ax^*)$$

and bound each term separately. Here we are defining $R_t = Z_t - t$, and $R_{a,x} = Z_{a,x} - a^T x$ and $R_{A,x} = Z_{A,x} - Ax$ similarly.

We present the proof of the following lemmas in Section J.7 and Section J.8 respectively.

**Lemma B.1.** *There is a constant $c = c(\alpha)$ such that under the conditions of Theorem A.7, with high probability over $(A, y)$, $\|\mathbb{E}[R_{A,\breve{x}}]\|_2^2 \leq cm$.*

**Lemma B.2.** *There is a constant $c_y = c_y(\alpha)$ such that $\|R_{A,x^*}\|_2^2 \le c_y m$ with high probability.*

Combining the above lemmas with the bound on $\|\breve{x} - x^*\|_2$ from the previous section, we get the desired theorem. See Section J.9 for the full proof.

**Theorem B.3.** *There are constants $M = M(\alpha)$, $\sigma = \sigma(\alpha)$, and $d = d(\alpha)$ with the following property. Suppose $m \ge Mk \log n$, and $\lambda = \sigma\sqrt{(\log n)/m}$. Then with high probability we have $\|\breve{z}_{U^c}\|_\infty < 1$.*

As an aside that we'll use later, this proof can be extended to any random vector near $\breve{x}$ with support contained in $U$ (proof in Section J.10).

**Theorem B.4.** *There are constants $M = M(\alpha)$, $\sigma = \sigma(\alpha)$, and $d = d(\alpha)$ with the following property. Suppose $m \ge Mk \log n$ and $\lambda = \sigma\sqrt{(\log n)/m}$. If $X \in \mathbb{R}^n$ is a random variable with $\operatorname{supp}(X) \subseteq U$ always, and $\|\breve{x} - X\|_2 \le 1/m$ with high probability, then with high probability $\left\|\frac{1}{m}A_{U^c}(\mathbb{E}Z_{A,X} - y)\right\|_\infty \le \lambda/2$.*

Returning to the goal of this section, it remains to show that $A_U^T A_U$ is invertible with high probability. But this follows from the isometry guarantee of Theorem G.1. Our main statistical result, Proposition 3.2, now follows.

**Proof of Proposition 3.2.** Take $M$, $\sigma$, and $d$ as in the statement of Theorem B.3. Let $m \ge Mk \log n$ and $\lambda = \sigma\sqrt{(\log n)/m}$. Let $\hat{x} \in \mathbb{R}^n$ be any optimal solution to the regularized program, and let $\breve{x} \in \mathbb{R}^U$ be any solution to the restricted program. By Theorem B.3, with high probability we have $\|x^* - \breve{x}\| \le d\sqrt{(k \log n)/m}$ and $\|\breve{z}_{U^c}\| < 1$; and by Theorem G.1, $A_U^T A_U$ is invertible. So by Lemma 5.2, it follows that $\breve{x} = \hat{x}$. Therefore $\|x^* - \hat{x}\| \le d\sqrt{(k \log n)/m}$. $\qquad\square$

## C  Primal-dual witness method

**Proof of Lemma 5.1.** For a single sample $(A_j, y_j)$, the partial derivative in direction $x_i$ is

$$
\frac{\partial}{\partial x_i} \operatorname{nll}(x; A_j, y_j) = A_{ji}(A_j x - y) + \frac{\frac{\partial}{\partial x_i}\int_S e^{-(A_j x - z)^2/2}\,dz}{\int_S e^{-(A_j x - z)^2/2}\,dz}
$$

$$
= A_{ji}(A_j x - y) - \frac{\int_S A_{ji}(A_j x - z)e^{-(A_j x - z)^2/2}\,dz}{\int_S e^{-(A_j x - z)^2/2}\,dz}
$$

$$
= A_{ji}(A_j x - y) - \mathbb{E}[A_{ji}(A_j x - Z_{A_j x})]
$$

where expectation is taken over the random variable $Z_{A_j x}$ (for fixed $A_j$). Simplifying yields the expression

$$
\nabla \operatorname{nll}(x; A_j, y_j) = A_j(\mathbb{E}[Z_{A_j x}] - y).
$$

The second partial derivative of $\operatorname{nll}(x; A_j, y_j)$ in directions $x_{i_1}$ and $x_{i_2}$ is therefore

$$
\frac{\partial^2}{\partial x_{i_1}\partial x_{i_2}} \operatorname{nll}(x; A_j, y_j) = \frac{\partial}{\partial x_{i_1}} A_{ji_2}(\mathbb{E}[Z_{A_j x}] - y)
$$

$$
= A_{ji_2}\frac{\partial}{\partial x_{i_1}}\left(\frac{\int_S ze^{-(A_j x - z)^2/2}\,dz}{\int_S e^{-(A_j x - z)^2/2}\,dz} - y\right)
$$

$$
= A_{ji_2}\left(\frac{\frac{\partial}{\partial x_{i_1}}\int_S ze^{-(A_j x - z)^2/2}\,dz}{\int_S e^{-(A_j x - z)^2/2}\,dz} - \right.
$$

$$
\left. \frac{\int_S ze^{-(A_j x - z)^2/2}\,dz \frac{\partial}{\partial x_{i_1}}\int_S e^{-(A_j x - z)^2/2}\,dz}{\left(\int_S e^{-(A_j x - z)^2/2}\,dz\right)^2}\right)
$$

$$
= A_{ji_2}(\mathbb{E}[-A_{ji_1}Z_{A_j x}(A_j x - Z_{A_j x})] - \mathbb{E}[Z_{A_j x}]\mathbb{E}[-A_{ji_1}(A_j x - Z_{A_j x})]
$$

$$
= A_{ji_1} A_{ji_2} \operatorname{Var}(Z_{A_j x}).
$$

We conclude that

$$H(x; A_j, y_j) = A_j^T A_j \operatorname{Var}(Z_{A_j x}).$$

Averaging over all samples yields the claimed result. $\qquad \square$

The following lemma collects several useful facts that are needed for the PDW method. Parts (a) and (b) are generically true for any $\ell_1$-regularized convex program; part (c) is a holdover from the untruncated setting that is still true. The proof is essentially due to [27], although part (c) now requires slightly more work.

**Lemma C.1.** *Fix any* $(A, y)$.

(a) *A vector* $x \in \mathbb{R}^n$ *is optimal for Program 2 if and only if there exists some* $z \in \partial \|x\|_1$ *such that*

$$\nabla \operatorname{nll}(x; A, y) + \lambda z = 0.$$

(b) *Suppose that* $(x, z)$ *are as in (a), and furthermore* $|z_i| < 1$ *for all* $i \notin \operatorname{supp}(x)$. *Then necessarily* $\operatorname{supp}(\hat{x}) \subseteq \operatorname{supp}(x)$ *for any optimal solution* $\hat{x}$ *to Program 2.*

(c) *Suppose that* $(x, z)$ *are as in (b), with* $I = \operatorname{supp}(x)$. *If* $A_I^T A_I$ *is invertible, then* $x$ *is the unique optimal solution to Program 2.*

*Proof.* Part (a) is simply the subgradient optimality condition in a convex program.

Part (b) is a standard fact about duality; we provide a proof here. Let $\hat{x}$ be any optimal solution to Program 2. We claim that $\hat{x}^T z = \|\hat{x}\|_1$. To see this, first note that $x^T z = \|x\|_1$, since $x_i z_i = |x_i|$ always holds by definition of a subgradient for the $\ell_1$ norm. Now, by optimality of $x$ and $\hat{x}$, we have $f(x) = f(\hat{x}) \le f(tx + (1-t)\hat{x})$ for all $0 \le t \le 1$. Therefore by convexity, $f(tx + (1-t)\hat{x}) = f(x)$ for all $0 \le t \le 1$. Since $f$ is the sum of two convex functions, both must be linear on the line segment between $x$ and $\hat{x}$. Therefore

$$\operatorname{nll}(tx + (1-t)\hat{x}) = t \operatorname{nll}(x) + (1-t)\operatorname{nll}(\hat{x})$$

for all $0 \le t \le 1$. We conclude that

$$(\nabla \operatorname{nll}(x)) \cdot (\hat{x} - x) = \operatorname{nll}(\hat{x}) - \operatorname{nll}(x) = \|x\|_1 - \|\hat{x}\|_1 .$$

Since $\nabla \operatorname{nll}(x) + z = 0$ by subgradient optimality, it follows that $z^T(\hat{x} - x) = \|\hat{x}\|_1 - \|x\|_1$. Hence, $z^T \hat{x} = \|\hat{x}\|_1$. Since $|z_i| \le 1$ for all $i$, if $|z_i| < 1$ for some $i$ then necessarily $\hat{x}_i = 0$ for equality to hold.

For part (c), if $A_I^T A_I$ is invertible, then it is (strictly) positive definite. The Hessian of Program 3 is

$$\frac{1}{m} \sum_{j=1}^m A_{I,j}^T A_{I,j} \operatorname{Var}(Z_{A_j,x}).$$

Since $\operatorname{Var}(Z_{A_j,x})$ is always positive, there is some $\epsilon > 0$ (not necessarily a constant) such that

$$\frac{1}{m} \sum_{j=1}^m A_{I,j}^T A_{I,j} \operatorname{Var}(Z_{A_j,x}) \succcurlyeq \frac{1}{m} \epsilon \sum_{j=1}^m A_{I,j}^T A_{I,j} = \frac{1}{m} \epsilon A_I^T A_I.$$

Thus, the Hessian of the restricted program is positive definite, so the restricted program is strictly convex. Therefore the restricted program has a unique solution. By part (b), any solution to the original program has support in $I$, so the original program also has a unique solution, which must be $x$. $\qquad \square$

As with the previous lemma, the following proof is essentially due to [27] (with a different subgradient optimality condition).

**Proof of Lemma 5.2.** By part (a) of Lemma C.1, a vector $x \in \mathbb{R}^n$ is optimal for Program 2 if and only if there is some $z \in \partial \|x\|_1$ such that

$$\frac{1}{m} A^T (\mathbb{E}Z_{A,x} - y) + \lambda z = 0.$$

This vector equality can be written in block form as follows:

$$\frac{1}{m} \begin{bmatrix} A_U^T \\ A_{U^c}^T \end{bmatrix} (\mathbb{E}Z_{A,x} - y) + \lambda \begin{bmatrix} z_U \\ z_{U^c} \end{bmatrix} = 0.$$

Since $\breve{x}$ is optimal in $\mathbb{R}^U$, there is some $\breve{z}_U \in \partial \|\breve{x}\|_1$ such that $(\breve{x}, \breve{z}_U)$ satisfy the first of the two block equations. This is precisely part (a). If furthermore $\breve{x}$ is zero-extended to $\mathbb{R}^n$, and $\breve{z}$ is extended as in part (b), and $\breve{z}$ satisfies $\|\breve{z}_{U^c}\|_\infty \leq 1$, then since $x_i = 0$ for all $i \notin U$, we have that $\breve{z}$ is a subgradient for $\|\breve{x}\|_1$. Therefore $\breve{x}$ is optimal for Program 2. If $\|\breve{z}_{U^c}\|_\infty < 1$ and $A_U^T A_U$ is invertible, then $\breve{x}$ is the unique solution to Program 2 by parts (b) and (c) of Lemma C.1. $\qquad\square$

## D  Sparse recovery from the Restricted Isometry Property

In this section we restate a theorem due to [6] about sparse recovery in the presence of noise. Our statement is slightly generalized to allow a trade-off between the isometry constants and the sparsity. That is, as the sparsity $k$ decreases relative to the isometry order $s$, the isometry constants $\tau, T$ are allowed to worsen.

**Theorem D.1** ([6])**.** *Let $B \in \mathbb{R}^{m \times n}$ be a matrix satisfying the s-Restricted Isometry Property*

$$\tau \|v\|_2 \leq \|Bv\|_2 \leq T \|v\|_2$$

*for all s-sparse $v \in \mathbb{R}^n$. Let $w^* \in \mathbb{R}^n$ be k-sparse for some $k < s$, and let $w \in \mathbb{R}^n$ satisfy $\|w\|_1 \leq \|w^*\|_1$. Then*

$$\|B(w - w^*)\|_2 \geq (\tau(1 - \rho) - T\rho) \|w - w^*\|_2$$

*where $\rho = \sqrt{k/(s - k)}$.*

*Proof.* Let $h = w - w^*$ and let $T_0 = \text{supp}(w^*)$. Then

$$\|w^*\|_1 \geq \|w\|_1 = \left\|h_{T_0^C}\right\|_1 + \|(h + w^*)_{T_0}\|_1 \geq \left\|h_{T_0^C}\right\|_1 + \|w^*\|_1 - \|h_{T_0}\|_1,$$

so $\|h_{T_0}\|_1 \geq \left\|h_{T_0^C}\right\|_1$. Without loss of generality assume that $T_0^C = \{1, \ldots, |T_0^C|\}$, and $|h_i| \geq |h_{i+1}|$ for all $1 \leq i < |T_0^C|$. Divide $T_0^C$ into sets of size $s' = s - k$ respecting this order:

$$T_0^C = T_1 \cup T_2 \cup \cdots \cup T_r.$$

Then the Restricted Isometry Property gives

$$\|Bh\|_2 \geq \|Bh_{T_0 \cup T_1}\|_2 - \sum_{t=2}^{r} \|Bh_{T_t}\|_2 \geq \tau \|h_{T_0 \cup T_1}\|_2 - T \sum_{t=2}^{r} \|h_{T_t}\|_2 \qquad (5)$$

For any $t \geq 1$ and $i \in T_{t+1}$, we have $h_i \leq \|h_{T_t}\|_1 / s'$, so that

$$\left\|h_{T_{t+1}}\right\|_2^2 \leq \frac{\|h_{T_t}\|_1^2}{s'}.$$

Summing over all $t \geq 2$, we get

$$\sum_{t=2}^{r} \|h_{T_t}\|_2 \leq \frac{1}{\sqrt{s'}} \sum_{t=1}^{r} \|h_{T_t}\|_1 = \frac{\left\|h_{T_0^C}\right\|_1}{\sqrt{s'}} \leq \frac{\|h_{T_0}\|_1}{\sqrt{s'}} \leq \sqrt{\frac{k}{s'}} \|h\|_2.$$

The triangle inequality implies that $\|h_{T_0 \cup T_1}\|_2 \geq (1 - \sqrt{k/s'}) \|h\|_2$. Returning to Equation 5, it follows that

$$\|Bh\|_2 \geq \left(\tau(1 - \sqrt{k/s'}) - T\sqrt{k/s'}\right) \|h\|_2$$

as claimed. $\qquad\square$

# E  Summary of the algorithm

---

**Algorithm 1** Projected Stochastic Gradient Descent.

---
1: **procedure** SGD($N, \lambda$)                 ▷ *N: number of steps, $\lambda$: parameter*
2:      $x^{(0)} \leftarrow \operatorname{argmin} \|x\|_1$ s.t. $x \in \mathscr{E}_r$          ▷ *see the Appendix F for details*
3:      **for** $t = 1, \ldots, N$ **do**
4:          $\eta_t \leftarrow \frac{1}{\sqrt{nN}}$
5:          $v^{(t)} \leftarrow$ GRADIENTESTIMATION($x^{(t-1)}$)
6:          $w^{(t)} \leftarrow x^{(t-1)} - \eta_t w^{(t)}$
7:          $x^{(t)} \leftarrow \operatorname{argmin}_{x \in \mathscr{E}_r} \left\| x - w^{(t)} \right\|_2$       ▷ *see the Appendix F for details*
8:      **return** $\bar{x} \leftarrow \frac{1}{N} \sum_{t=1}^{N} x^{(t)}$           ▷ *output the average*

---

---

**Algorithm 2** The function to estimate the gradient of the $\ell_1$ regularized negative log-likelihood.

---
1: **function** GRADIENTESTIMATION($x$)
2:      Pick $j$ at random from $[n]$
3:      Use Assumption II or Lemma K.4 to sample $z \sim Z_{A_j x^{(t)}}$
4:      **return** $A_j(z - y_j)$

---

# F  Algorithm details

In this section we fill in the missing details about the algorithm's efficiency. Since we have already seen that the algorithm converges in $O(\mathrm{poly}(n))$ update steps, all that remains is to show that the following algorithmic problems can be solved efficiently:

1. (Initial point) Compute $x^{(0)} = \operatorname{argmin}_{x \in \mathscr{E}_r} \|x\|_1$.

2. (Stochastic gradient) Given $x^{(t)} \in \mathscr{E}_r$ and $j \in [m]$, compute a sample $A_j(z - y_j)$, where $z \sim Z_{A_j x^{(t)}}$.

3. (Projection) Given $w^{(t)} \in \mathbb{R}^n$, compute $x^{(t+1)} = \operatorname{argmin}_{x \in \mathscr{E}_r} \left\| x - w^{(t)} \right\|_2$.

**Initial point.** To obtain the initial point $x^{(0)}$, we need to solve the program

$$
\begin{aligned}
\text{minimize} \quad & \|x\|_1 \\
\text{subject to} \quad & \|Ax - y\|_2 \leq r\sqrt{m}.
\end{aligned}
$$

This program has come up previously in the compressed sensing literature (see, e.g., [6]). It can be recast as a Second-Order Cone Program (SOCP) by introducing variables $x^+, x^- \in \mathbb{R}^n$:

$$
\begin{aligned}
\text{minimize} \quad & \sum_{i=1}^{n}(x_i^+ - x_i^-) \\
\text{subject to} \quad & \|Ax^+ - Ax^- - y\|_2 \leq r\sqrt{m}, \\
& x^+ \geq 0, \\
& -x^- \geq 0.
\end{aligned}
$$

Thus, it can be solved in polynomial time by interior-point methods (see [4]).

**Stochastic gradient.** In computing an unbiased estimate of the gradient, the only challenge is sampling from $Z_{A_j x^{(t)}}$. By Assumption II, this takes $T(\gamma_S(A_j x^{(t)}))$ time. We know from Lemma I.4 that $\gamma_S(A_j x^*) \geq \alpha^{2m}$. Since $x^{(t)}, x^* \in \mathscr{E}_r$, we have from Lemma I.2 that

$$
\gamma_S(A_j x^{(t)}) \geq \gamma_S(t^*)^2 e^{-|A_j(x^{(t)} - x^*)|^2 - 2} \geq \alpha^{4m} e^{-4r^2 m - 2} \geq e^{-\Theta(m/\alpha)}.
$$

Thus, the time complexity of computing the stochastic gradient is $T(e^{-\Theta(m/\alpha)})$.

In the special case when the truncation set $S$ is a union of $r$ intervals, there is a sampling algorithm with time complexity $T(\beta) = \text{poly}(r, \log(1/\beta, n))$ (Lemma K.4). Hence, in this case the time complexity of computing the stochastic gradient is $\text{poly}(r, n)$.

To be more precise, we instantiate Lemma K.4 with accuracy $\zeta = 1/(nL)$, where $L = \text{poly}(n)$ is the number of update steps performed. This gives some sampling algorithm $\mathscr{A}$. In each step, $\mathscr{A}$'s output distribution is within $\zeta$ of the true distribution $N(t, 1; S)$. Consider a hypothetical sampling algorithm $\mathscr{A}'$ in which $\mathscr{A}$ is run, and then the output is altered by rejection to match the true distribution. Alteration occurs with probability $\zeta$. Thus, running the PSGD algorithm with $\mathscr{A}'$, the probability that any alteration occurs is at most $L\zeta = o(1)$. As shown by Theorem H.1, PSGD with $\mathscr{A}'$ succeeds with high probability. Hence, PSGD with $\mathscr{A}$ succeeds with high probability as well.

**Projection.**  The other problem we need to solve is projection onto set $\mathscr{E}_r$:

$$
\begin{array}{ll}
\text{minimize} & \|x - v\|_2 \\
\text{subject to} & \|Ax - y\|_2 \le r\sqrt{m}.
\end{array}
$$

This is a convex QCQP, and therefore solvable in polynomial time by interior point methods (see [4]).

## G   Isometry properties

Let $A \in \mathbb{R}^{m \times n}$ consist of $m$ samples $A_i$ from Process 1. In this section we prove the following theorem:

**Theorem G.1.** *For every $\epsilon > 0$ there are constants $\delta > 0$, $M$, $\tau > 0$ and $T$ with the following property. Let $V \subseteq [n]$. Suppose that $m \ge M|V|$. With probability at least $1 - e^{-\delta m}$ over $A$, for every subset $J \subseteq [m]$ with $|J| \ge \epsilon m$, the $|J| \times k$ submatrix $A_{J,V}$ satisfies*

$$
\tau\sqrt{m}\,\|v\|_2 \le \|A_{J,V}\,v\|_2 \le T\sqrt{m}\,\|v\|_2 \qquad \forall\, v \in \mathbb{R}^V.
$$

We start with the upper bound, for which it suffices to take $J = [m]$.

**Lemma G.2.** *Let $V \subseteq [n]$. Suppose that $m \ge |V|$. There is a constant $T = T(\alpha)$ such that*

$$
\Pr[s_{\max}(A_V) > T] \le e^{-\Omega(m)}.
$$

*Proof.*  In the process for generating $A$, consider the matrix $A'$ obtained by not discarding any of the samples $a \in \mathbb{R}^n$. Then $A'$ is a $\xi \times n$ matrix for a random variable $\xi$; each row of $A'$ is a spherical Gaussian independent of all previous rows, but $\xi$ depends on the rows. Nonetheless, by a Chernoff bound, $\Pr[\xi > 2m/\alpha] \le e^{-m/(3\alpha)}$. In this event, $A'$ is a submatrix of $2m/\alpha \times n$ matrix $B$ with i.i.d. Gaussian entries. By [22],

$$
\Pr[s_{\max}(B_V) > C\sqrt{2m/\alpha}] \le e^{-cm}
$$

for some absolute constants $c, C > 0$. Since $A'$ is a submatrix of $B$ with high probability, and $A$ is a submatrix of $A'$, it follows that

$$
\Pr[s_{\max}(A_V) > C\sqrt{2m/\alpha}] \le e^{-\Omega(m)}
$$

as desired.  $\square$

For the lower bound, we use an $\epsilon$-net argument.

**Lemma G.3.** *Let $\epsilon > 0$ and let $v \in \mathbb{R}^n$ with $\|v\|_2 = 1$. Let $a \sim N(0,1)^n$. Then*

$$
\Pr[|a^T v| < \alpha\epsilon\sqrt{\pi/2}|a^T x^* + Z \in S] < \epsilon.
$$

*Proof.*  From the constant survival probability assumption,

$$
\Pr[|a^T v| < \delta|a^T x^* + Z \in S] \le \alpha^{-1}\Pr[|a^T v| < \delta].
$$

But $a^T v \sim N(0,1)$, so $\Pr[|a^T v| < \delta] \le 2\delta/\sqrt{2\pi}$. Taking $\delta = \alpha\epsilon\sqrt{\pi/2}$ yields the desired bound.  $\square$

**Lemma G.4.** *Let $V \subseteq [n]$. Fix $\epsilon > 0$ and fix $v \in \mathbb{R}^V$ with $\|v\|_2 = 1$. There are positive constants $\tau_0 = \tau_0(\alpha, \epsilon)$ and $c_0 = c_0(\alpha, \epsilon)$ such that*

$$\Pr\left[\exists J \subseteq [m] : (|J| \geq \epsilon m) \wedge (\|A_{J,V} v\|_2 < \tau_0)\right] \leq e^{-c_0 m}.$$

*Proof.* For each $j \in [m]$ let $B_j$ be the indicator random variable for the event that $|A_{j,V} v| < \alpha\epsilon/3$. Let $B = \sum_{j=1}^{m} B_j$. By Lemma G.3, $\mathbb{E}B < \epsilon m/3$. Each $B_j$ is independent, so by a Chernoff bound,

$$\Pr[B > \epsilon m/2] \leq e^{-\epsilon m/18}.$$

In the event $[B \leq \epsilon m/2]$, for any $J \subseteq [m]$ with $|J| \geq \epsilon m$ it holds that

$$\|A_{J,V} v\|_2^2 = \sum_{j \in J} (A_{j,V} v)^2 \geq \sum_{j \in J : B_j = 0} (A_{j,V} v)^2 \geq (\alpha\epsilon/3)B \geq \alpha\epsilon^2 m/6.$$

So the event in the lemma statement occurs with probability at most $e^{-\epsilon m/18}$. $\qquad\square$

Now we can prove the isometry property claimed in Theorem G.1.

**Proof of Theorem G.1.** Let $V \subseteq [n]$. Let $\epsilon > 0$. Take $\gamma = 4|V|/(c_0 m)$, where $c_0 = c_0(\alpha, \epsilon)$ is the constant in the statement of Lemma G.4. Let $\mathscr{B} \subseteq \mathbb{R}^V$ be the $k$-dimensional unit ball. Let $\mathscr{D} \subset \mathscr{B}$ be a maximal packing of $(1 + \gamma/2)\mathscr{B}$ by radius-$(\gamma/2)$ balls with centers on the unit sphere. By a volume argument,

$$|\mathscr{D}| \leq \frac{(1 + \gamma/2)^k}{(\gamma/2)^k} \leq e^{2k/\gamma} \leq e^{c_0 m/2}.$$

Applying Lemma G.4 to each $v \in \mathscr{D}$ and taking a union bound,

$$\Pr[\exists J \subseteq [m], v \in \mathscr{D} : (|J| \geq \epsilon m) \wedge (\|A_{J,V} v\|_2 < \tau_0)] \leq e^{-c_0 m/2}.$$

So with probability $1 - e^{-\Omega(m)}$, the complement of this event holds. And by Lemma G.2, the event $s_{\max}(A_V) \leq T\sqrt{m}$ holds with probability $1 - e^{-\Omega(m)}$. In these events we claim that the conclusion of the theorem holds. Take any $v \in \mathbb{R}^V$ with $\|v\|_2 = 1$, and take any $J \subseteq [m]$ with $|J| \geq \epsilon m$. Since $\mathscr{D}$ is maximal, there is some $w \in \mathscr{D}$ with $\|v - w\|_2 \leq \gamma$. Then

$$\|A_{J,V} v\|_2 \geq \|A_{J,V} w\|_2 - \|A_{J,V}(v - w)\|_2 \geq \tau_0 - \gamma T.$$

But $\gamma \leq 4/(c_0 M)$. For sufficiently large $M$, we get $\gamma < \tau_0/(2T)$. Taking $\tau = \tau_0/2$ yields the claimed lower bound. $\qquad\square$

As a corollary, we get that $A_U^T$ is a $\sqrt{m}$-isometry on its row space up to constants (of course, this holds for any $V \subseteq [n]$ with $|V| = k$, but we only need it for $V = U$).

**Corollary G.5.** *With high probability, for every $u \in \mathbb{R}^k$,*

$$\frac{\tau^2}{T}\sqrt{m}\,\|A_U u\|_2 \leq \|A_U^T A_U u\|_2 \leq \frac{T^2}{\tau}\sqrt{m}\,\|A_U u\|_2.$$

*Proof.* By Theorem G.1, with high probability all eigenvalues of $A_U^T A_U$ lie in the interval $[\tau\sqrt{m}, T\sqrt{m}]$. Hence, all eigenvalues of $(A_U^T A_U)^2$ lie in the interval $[\tau^2 m, T^2 m]$. But then

$$\|A_U^T A_U u\|_2 = u^T (A_U^T A_U)^2 u \geq \tau^2 m u^T u \geq \frac{\tau^2}{T}\sqrt{m}\,\|A_U u\|_2.$$

The upper bound is similar. $\qquad\square$

We also get a Restricted Isometry Property, by applying Theorem G.1 to all subsets $V \subseteq [n]$ of a fixed size.

**Corollary G.6** (Restricted Isometry Property)**.** *There is a constant $M$ such that for any $s > 0$, if $m \geq Ms \log n$, then with high probability, for every $v \in \mathbb{R}^n$ with $|\operatorname{supp}(v)| \leq s$,*

$$\tau\sqrt{m}\,\|v\|_2 \leq \|Av\|_2 \leq T\sqrt{m}\,\|v\|_2.$$

*Proof.* We apply Theorem G.1 to all $V \subseteq [n]$ with $|V| = s$, and take a union bound over the respective failure events. The probability that there exists some set $V \subseteq [n]$ of size $s$ such that the isometry fails is at most

$$\binom{n}{s} e^{-\delta m} \leq e^{s \log n - \delta m}.$$

If $m \geq Ms \log n$ for a sufficiently large constant $M$, then this probability is $o(1)$. $\qquad\square$

From this corollary, our main result for adversarial noise (Theorem 3.4) follows almost immediately:

**Proof of Theorem 3.4.** Let $M'$ be the constant in Corollary G.6. Let $\rho = \min(\tau/(4T), 1/3)$, and let $M = (1 + 1/\rho^2)M'$. Finally, let $s = (1 + 1/\rho^2)k$.

Let $\epsilon > 0$. Suppose that $m \geq Mk \log n$ and $\|Ax^* - y\| \leq \epsilon$. Then $m \geq M's \log n$, so by Corollary G.6, $A/\sqrt{m}$ satisfies the $s$-Restricted Isometry Property.

By definition, $\hat{x}$ satisfies $\|A\hat{x} - y\|_2 \leq \epsilon$ and $\|\hat{x}\|_1 \leq \|x^*\|_1$ (by feasibility of $x^*$). Finally, $x^*$ is $k$-sparse. We conclude from Theorem D.1 and our choice of $\rho$ that

$$\left\|(A/\sqrt{m})(\hat{x} - x^*)\right\|_2 \geq (\tau(1 - \rho) - T\rho) \|\hat{x} - x^*\|_2 \geq \frac{\tau}{2} \|\hat{x} - x^*\|_2.$$

But $\|A(\hat{x} - x^*)\|_2 \leq 2\epsilon$ by the triangle inequality. Thus, $\|\hat{x} - x^*\|_2 \leq \tau\epsilon/\sqrt{m}$. $\qquad\square$

## H  Projected Stochastic Gradient Descent

In this section we present the exact PSGD convergence theorem which we use, together with a proof for completeness.

**Theorem H.1.** *Let $f : \mathbb{R}^n \to \mathbb{R}$ be a convex function achieving its optimum at $\check{x} \in \mathbb{R}^n$. Let $\mathscr{P} \subseteq \mathbb{R}^n$ be a convex set containing $\check{x}$. Let $x^{(0)} \in \mathscr{P}$ be arbitrary. For $1 \leq t \leq T$ define a random variable $x^{(t)}$ by*

$$x^{(t)} = Proj_{\mathscr{P}}(x^{(t-1)} - \eta v^{(t-1)}),$$

*where $\mathbb{E}[v^{(t)}|x^{(t)}] \in \partial f(x^{(t)})$ and $\eta$ is fixed. Then*

$$\mathbb{E}[f(\bar{x})] - f(\check{x}) \leq (\eta T)^{-1}\mathbb{E}\left[\left\|x^{(0)} - \check{x}\right\|_2^2\right] + \eta T^{-1}\sum_{i=1}^{T}\mathbb{E}\left[\left\|v^{(i)}\right\|_2^2\right]$$

*where $\bar{x} = \frac{1}{T}\sum_{i=1}^{T} x^{(i)}$.*

*Proof.* Fix $0 \leq t < T$. We can write

$$\left\|x^{(t+1)} - \check{x}\right\|_2^2 \leq \left\|(x^{(t)} - \eta v^{(t)}) - \check{x}\right\|_2^2 = \left\|x^{(t)} - \check{x}\right\|_2^2 - 2\eta\langle v^{(t)}, x^{(t)} - \check{x}\rangle + \eta^2\left\|v^{(t)}\right\|_2^2$$

since projecting onto $\mathscr{E}_r$ cannot increase the distance to $\check{x} \in \mathscr{E}_r$.

Taking expectation over $v^{(t)}$ for fixed $x^{(0)}, \ldots, x^{(k)}$, we have

$$\mathbb{E}\left[\left\|x^{(t+1)} - \check{x}\right\|_2^2 \Big| x^{(0)}, \ldots, x^{(k)}\right] \leq \left\|x^{(t)} - \check{x}\right\|_2^2 - 2\eta\langle \mathbb{E}v^{(t)}, x^{(t)} - \check{x}\rangle + \eta^2\mathbb{E}\left[\left\|v^{(t)}\right\|_2^2\right]$$

$$\leq \left\|x^{(t)} - \check{x}\right\|_2^2 - 2\eta(f(x^{(t)}) - f(\check{x})) + \eta^2\mathbb{E}\left[\left\|v^{(t)}\right\|_2^2\right]$$

where the last inequality uses the fact that $\mathbb{E}v^{(t)}$ is a subgradient for $f$ at $x^{(t)}$. Rearranging and taking expectation over $x^{(0)}, \ldots, x^{(t)}$, we get that

$$2\left(\mathbb{E}\left[f(x^{(t)})\right] - f(\check{x})\right) \leq \eta^{-1}\left(\mathbb{E}\left[\left\|x^{(t)} - \check{x}\right\|_2^2\right] - \mathbb{E}\left[\left\|x^{(t+1)} - \check{x}\right\|_2^2\right]\right) + \eta\mathbb{E}\left[\left\|v^{(t)}\right\|_2^2\right].$$

But now summing over $0 \le t < T$, the right-hand side of the inequality telescopes, giving

$$\mathbb{E}[f(\bar{x})] - f(\check{x}) \le \frac{1}{T} \sum_{t=0}^{T-1} \mathbb{E}[f(x^{(t)})] - f(\check{x})$$

$$\le \frac{1}{\eta T} \mathbb{E}\left[\left\|x^{(0)} - \check{x}\right\|_2^2\right] + \frac{\eta}{T} \sum_{t=0}^{T-1} \mathbb{E}\left[\left\|v^{(t)}\right\|_2^2\right].$$

This is the desired bound. $\qquad\square$

## I   Survival probability

In this section we collect useful lemmas about truncated Gaussian random variables and survival probabilities.

**Lemma I.1** ([9]). *Let $t \in \mathbb{R}$ and let $S \subset \mathbb{R}$ be a measurable set. Then $\mathrm{Var}(Z_t) \ge C\gamma_S(t)^2$ for a constant $C > 0$.*

**Lemma I.2** ([9]). *For $t, t^* \in \mathbb{R}$,*

$$\log \frac{1}{\gamma_S(t)} \le 2 \log \frac{1}{\gamma_S(t^*)} + |t - t^*|^2 + 2.$$

**Lemma I.3** ([9]). *For $t \in \mathbb{R}$,*

$$\mathbb{E}[R_t^2] \le 2 \log \frac{1}{\gamma_S(t)} + 4.$$

**Lemma I.4.** *With high probability,*

$$\sum_{j=1}^{m} \log \frac{1}{\gamma_S(A_j x^*)} \le 2m \log\left(\frac{1}{\alpha}\right).$$

*Proof.* Let $X_j = \log 1/\gamma_S(A_j x^*)$ for $j \in [m]$, and let $X = X_1 + \cdots + X_m$. Since $X_1, \ldots, X_m$ are independent and identically distributed,

$$\mathbb{E}[e^X] = \mathbb{E}[e^{X_j}]^m = \mathbb{E}\left[\frac{1}{\gamma_S(A_j x^*)}\right]^m = \left(\frac{\mathbb{E}_{a \sim N(0,1)^n}[1]}{\mathbb{E}_{a \sim N(0,1)^n}[\gamma_S(a^T x^*)]}\right)^m \le \alpha^{-m}.$$

Therefore

$$\Pr[X > 2m \log 1/\alpha] = \Pr[e^X > e^{2m \log 1/\alpha}] \le e^{-m \log 1/\alpha}$$

by Markov's inequality. $\qquad\square$

## J   Omitted proofs

### J.1   Proof of Lemma A.1

We start by bounding the subGaussian variance proxy of a truncated Gaussian random variable. The following tail bound is standard:

**Lemma J.1** (Gaussian tail lower bound). *Let $Z \sim N(0,1)$. For any $t > 0$,*

$$\phi(t) := \Pr(Z > t) \ge \frac{e^{-t^2/2}}{C(t+1)}$$

*for an absolute constant $C > 0$.*

**Lemma J.2** (Moments of tail-truncated Gaussian). *Let $p \ge 1$ be integer, and let $t > 0$. Then*

$$F(t) := \phi(t)^{-1} \int_t^\infty x^p e^{-x^2/2} \, dx \le (C_2(t+1)\sqrt{p})^p.$$

*Proof.* We distinguish two cases. Suppose $t \leq 2\sqrt{p}$. Then

$$F(t) \leq \phi(t)^{-1} \int_{-\infty}^{\infty} |x|^p e^{-x^2/2}\, dx \leq C(t+1)e^{t^2/2} p^{p/2}$$

by moment bounds for the standard normal. But $t + 1 \leq e^t \leq e^{2\sqrt{p}}$ and $e^{t^2/2} \leq e^{2p}$, so

$$F(t) \leq C(e^4 \sqrt{p})^p.$$

Now conversely suppose $t > 2\sqrt{p}$. Define $f(x) = x^p e^{-x^2/2}$. For any $x \geq t$ and $\delta > 0$, we have

$$\frac{f(x+\delta)}{f(x)} = \left(1 + \frac{\delta}{x}\right)^p e^{-(\delta x + \delta^2/2)} \leq \exp\left(\frac{p\delta}{x} - \delta x - \delta^2/2\right) \leq \exp(-\delta t/2 - \delta^2/2).$$

Therefore taking $\delta = 1/(t+1)$, the decay is $\exp(-t/(2(t+1)) - 1/(2(t+1)^2)) \leq \exp(-1/3)$, so

$$\int_t^{\infty} f(x)\, dx \leq \sum_{k=0}^{\infty} (t+1)^{-1} \int_{t+k/(t+1)}^{t+(k+1)/(t+1)} f(x)\, dx \leq \sum_{k=0}^{\infty} (t+1)^{-1} f(t) e^{-k/3} \leq 4(t+1)^{-1} f(t).$$

So

$$F(t) \leq 4\phi(t)^{-1}(t+1)^{-1} f(t) \leq 4Ct^p$$

as desired. $\qquad\square$

**Lemma J.3** (Moments of truncated Gaussian). *Let $p \geq 1$ be integer, and let $S \subseteq \mathbb{R}$. Let $\alpha = \Pr[Z \in S]$ and let $X \sim Z|(Z \in S)$. Then*

$$(\mathbb{E}|X|^p)^{1/p} \leq C_3 \sqrt{p \log(2/\alpha)}.$$

*Proof.* Observe that $\mathbb{E}|X|^p = \mathbb{E}[|Z|^p | Z \in S] = \alpha^{-1} \mathbb{E}[|Z|^p \mathbb{1}_{Z \in S}]$. For fixed $\alpha$, this expectation is maximized when $S = (-\infty, a] \cup [b, \infty)$ for some $a < 0 < b$ (shifting mass towards the tails only increases the expectation). In fact, it's maximized when $a = -b$. In this case we have $\alpha = 2\phi(b)$ and

$$\mathbb{E}[|Z|^p \mathbb{1}_{|Z|>b}] = 2 \int_b^{\infty} x^p e^{-x^2/2}\, dx \leq 2\phi(b)(C_2 b \sqrt{p})^p.$$

As a consequence,

$$\mathbb{E}|X|^p \leq (C_2(b+1)\sqrt{p})^p.$$

But $\alpha = 2\phi(b) \leq 2e^{-b^2/2}$. So in terms of $\alpha$, the moment is bounded as

$$(\mathbb{E}|X|^p)^{1/p} \leq C_2 \sqrt{p}(1 + \sqrt{2\log(2/\alpha)})$$

as claimed. $\qquad\square$

**Corollary J.4** (Truncated Gaussian is subGaussian). *Let $S \subseteq \mathbb{R}$. Let $\alpha = \Pr[Z \in S]$ and let $X \sim Z|Z \in S$. Then*

$$X - \mathbb{E}X \sim \mathrm{subG}(C_4 \log(2/\alpha)).$$

*Proof.* Let $Y = X - \mathbb{E}X$. By the previous lemma, $|\mathbb{E}X| \leq C_3 \sqrt{\log(2/\alpha)}$ and for any integer $p \geq 1$,

$$(\mathbb{E}|Y|^p)^{1/p} \leq (\mathbb{E}|X|^p)^{1/p} + |\mathbb{E}X| \leq 2C_3 \sqrt{p \log(2/\alpha)}.$$

Since $Y$ has mean 0, it follows from subGaussian equivalencies that $Y \sim \mathrm{subG}(C_4 \log(2/\alpha))$. $\quad\square$

Now we can prove Lemma A.1.

**Proof of Lemma A.1.**

Recall that $\mathbb{E}\gamma_S(a^T x^*) \geq \alpha > 0$ for $a \sim N(0,1)^n$, and $(A, y)$ are the $m$ truncated linear samples of $x^*$.

Fix $i \in U$. For any $j \in [m]$, we have

$$\mathbb{E}[\gamma_S(A_j x^*)^{-1}] = \frac{\mathbb{E}_{a \sim N(0,1)^n}[\gamma_S(a^T x^*)^{-1}\gamma_S(a^T x^*)]}{\mathbb{E}_{a \sim N(0,1)^n}[\gamma_S(a^T x^*)]} \leq \alpha^{-1},$$

so for $t > 0$ we get the tail bound

$$\Pr[\log(2/\gamma_S(A_j x^*)) > t] \leq 2\alpha^{-1}e^{-t}.$$

Since $A_{ji}$ is the result of conditioning a Gaussian on an event of likelihood at least $\alpha$, we also get the tail bound

$$\Pr[A_{ji}^2 > t] \leq 2\alpha^{-1}e^{-t/2}.$$

So

$$\Pr[A_{ji}^2 \log(2/\gamma_S(A_j x^*)) > t] \leq 4\alpha^{-1}e^{-\sqrt{t}/2}.$$

By concentration of sums of independent, heavy-tailed random variables (see, e.g. Theorem 8.4 in [18]), it follows that with probability $1 - e^{-\Omega(m^{1/5})}$,

$$\sum_{j \in [m]} A_{ji}^2 \log(2/\gamma_S(A_j x^*)) \leq O(m).$$

With probability $1 - ke^{-\Omega(m^{1/5})}$, this above bound holds for all $i \in U$. Call this event $E$.

Condition on $A$ and suppose that event $E$ occurs. For fixed $A$, we can write $y_j \sim N(A_j x^*, 1, S)$ for $j \in [m]$. Moreover

$$y_j - \mathbb{E}y_j \sim \text{subG}(C_4 \log(2/\gamma_S(A_j x^*))).$$

Fix $i \in U$. Then $(A^T)_i \cdot (y - \mathbb{E}y)$ is subGaussian with variance proxy $C_4 \sum_{j \in [m]} A_{ji}^2 \log(2/\gamma_S(A_j x^*))$. Since event $E$ is assumed to hold, we have that $(A^T)_i \cdot (y - \mathbb{E}y)$ is subGaussian with variance proxy $C_5 m$. So

$$\Pr_y[|(A^T)_i \cdot (y - \mathbb{E}y)| > \sqrt{10 C_5 m \log n}] \leq \frac{1}{n^{10}}.$$

Thus, union bounding and summing over $i \in U$, we have that

$$\left\| A_U^T(y - \mathbb{E}y) \right\|_2^2 \leq C_6 mk \log n$$

with probability $1 - 1/n^9$ over $y$. Taking probability over $A$ and accounting for the probability that event $E$ fails, the total failure probability is $O(1/n^9 + ke^{-\Omega(m^{-1/5})})$. $\qquad\square$

## J.2  Proof of Lemma A.2

We can write

$$\mu_t = \frac{\int_S x e^{-(x-t)^2/2}\, dx}{\int_S e^{-(x-t)^2/2}\, dx}.$$

By the quotient rule,

$$\begin{aligned}
\frac{d}{dt}\mu_t &= -\frac{\int_S x(t-x)e^{-(x-t)^2/2}\, dx}{\int_S e^{-(x-t)^2/2}\, dx} + \frac{\left(\int_S x e^{-(x-t)^2/2}\, dx\right)\left(\int_S (t-x)e^{-(x-t)^2/2}\, dx\right)}{\left(\int_S e^{-(x-t)^2/2}\, dx\right)^2} \\
&= -\mathbb{E}[Z_t(t - Z_t)] + \mathbb{E}[Z_t]\mathbb{E}[t - Z_t] \\
&= \text{Var}(Z_t)
\end{aligned}$$

as desired.

### J.3 Proof of Lemma A.3

The fact that $\text{sign}(\mu_t - \mu_{t^*}) = \text{sign}(t - t^*)$ follows immediately from the fact that $\frac{d}{dt}\mu_t = \text{Var}(Z_t) \geq 0$ (Lemma A.2).

We now prove the second claim of the lemma. Suppose $t^* < t$; the other case is symmetric. Then we have

$$\mu_t - \mu_{t^*} = \int_{t^*}^t \text{Var}(Z_r)\, dr \geq C \int_{t^*}^t \gamma_S(r)^2\, dr \geq C\beta^2 \int_0^{t-t^*} e^{-r^2-2}\, dr$$

by Lemmas A.2, I.1 and I.2 respectively. But we can lower bound

$$\int_0^{t-t^*} e^{-r^2-2}\, dr \geq \int_0^{\min(1, t-t^*)} e^{-r^2-2}\, dr$$
$$\geq e^{-3}\min(1, t-t^*).$$

This bound has the desired form.

### J.4 Proof of Lemma A.4

Since $\mathbb{E}_{a \sim N(0,1)^k}\gamma_S(a^T x^*) \geq \alpha$ and $\gamma_S(a^T x^*)$ is always at most 1, we have $\Pr[\gamma_S(a^T x^*) \leq \alpha/2] \leq 1 - \alpha/2$. Since the samples are rejection sampled on $\gamma_S(a^T x^*)$, it follows that $\Pr[\gamma_S(A_j x^*) \leq \alpha/2] \leq 1 - \alpha/2$ as well. So by a Chernoff bound, with high probability, the number of $j \in [m]$ such that $\gamma_S(A_j x^*) \leq \alpha/2$ is at most $(1 - \alpha/3)m$.

The condition that $|A_j x^* - A_j \breve{x}|^2 \geq (6/(\alpha m))\|Ax^* - A\breve{x}\|^2$ is clearly satisfied by at most $(\alpha/6)m$ indices.

### J.5 Proof of Lemma A.5

By Lemma A.4 and Theorem G.1, with high probability $A_{I_{\text{good}}, U}$ and $A_U$ both have singular values bounded between $\sqrt{\tau m}$ and $\sqrt{Tm}$ for some positive constants $\tau = \tau(\alpha)$ and $T = T(\alpha)$. In this event, we have

$$\|A(x^* - \breve{x})\|_{I_{\text{good}}}^2 \geq \tau m \|x^* - \breve{x}\|^2 \geq \frac{\tau}{T}\|A(x^* - \breve{x})\|^2$$

which proves the claim.

### J.6 Proof of Theorems A.6 and A.7

**Proof of Theorem A.6.** Let $a = A(\breve{x} - x^*)$ and let $b = \mu_{A\breve{x}} - \mu_{Ax^*}$. Our aim is to show that if $\|a\|_2$ is large, then $\|A_U^T b\|_2$ is large, which would contradict Equation 4. Since $A_U^T$ is not an isometry, we can't simply show that $\|b\|_2$ is large. Instead, we write an orthogonal decomposition $b = v + A_U u$ for some $u \in \mathbb{R}^k$ and $v \in \mathbb{R}^m$ with $A_U^T v = 0$. We'll show that $\|A_U u\|_2$ is large. Since $A_U^T b = A_U^T A_U u$, and $A_U^T$ is an isometry on the row space of $A_U$, this suffices.

For every $j \in I_{\text{good}}$ with $|a_j| > 0$, we have by Lemma A.3 that

$$|b_j| \geq C\min(1, |a_j|) = C|a_j|\min(1/|a_j|, 1)$$

where $C$ is the constant which makes Lemma A.3 work for indices $j$ with $\gamma_S(A_j x^*) \geq \alpha/2$. Take $C' = \sqrt{6/\alpha}$, and suppose that the theorem's conclusion is false, i.e. $\|a\|_2 > C'\sqrt{m}$. Also suppose that the events of Lemmas A.1 and A.5 hold.

Then by the bound $|a_j|^2 \leq (6/(\alpha m))\|a\|_2^2$ for $j \in I_{\text{good}}$ we get

$$|b_j| \geq C|a_j|\min\left(\frac{\sqrt{\alpha/6}\sqrt{m}}{\|a\|_2}, 1\right) = \frac{c\sqrt{m}}{\|a\|_2}|a_j| \tag{6}$$

where $c = C\sqrt{\alpha/6}$. We assumed earlier that $|a_j| > 0$ but Equation 6 certainly also holds when $|a_j| = 0$.

By Lemma A.3, $a_j$ and $b_j$ have the same sign for all $j \in [m]$. So $a_j b_j \geq 0$ for all $j \in [m]$. Moreover, together with Equation 6, the sign constraint implies that for $j \in I_{\text{good}}$,

$$a_j b_j \geq \frac{c\sqrt{m}}{\|a\|} a_j^2.$$

Summing over $j \in I_{\text{good}}$ we get

$$\sum_{j \in I_{\text{good}}} a_j^2 \leq \frac{\|a\|_2}{c\sqrt{m}} \sum_{j \in I_{\text{good}}} a_j b_j \leq \frac{\|a\|_2}{c\sqrt{m}} \langle a, b \rangle = \frac{\|a\|_2}{c\sqrt{m}} \langle a, A_U u \rangle.$$

By Lemma A.5 on the LHS and Cauchy-Schwarz on the RHS, we get

$$\epsilon \|a\|_2^2 \leq \frac{\|a\|_2^2}{c\sqrt{m}} \|A_U u\|_2.$$

Hence $\|A_U u\|_2 \geq \epsilon c \sqrt{m}$. But then $\|A_U^T b\|_2 = \|A_U^T A_U u\|_2 \geq (\tau^2/T)\epsilon cm$. On the other hand, Equation 4 implies that

$$\frac{1}{m} \|A_U^T b\|_2 \leq \lambda \sqrt{k} + \frac{1}{m} \|A_U^T (\mathbb{E}[Z_{A,x^*}] - y)\|_2 \leq c'_{\text{reg}} + \sqrt{\alpha^{-1}(k \log n)/m}$$

since event (2) holds. This is a contradiction for $M'$ sufficiently large and $c'_{\text{reg}}$ sufficiently small. So either the assumption $\|a\|_2 > C'\sqrt{m}$ is false, or the events of Lemma A.1 or A.5 fail. But the latter two events fail with probability $o(1)$. So $\|a\|_2 \leq C'\sqrt{m}$ with high probability.

$\square$

Now that we know that $\|x^* - \breve{x}\|_2 \leq O(1)$, we can bootstrap to show that $\|x^* - \breve{x}\|_2 \leq \sqrt{(k \log n)/m}$. While the previous proof relied on the constant regime of the lower bound provided by Lemma A.3, the following proof relies on the linear regime.

**Proof of Theorem A.7.** As before, let $a = A(\breve{x} - x^*)$ and $b = \mu_{A\breve{x}} - \mu_{Ax^*}$. Suppose that the conclusion of Theorem A.6 holds, i.e. $\|a\|_2 \leq C'\sqrt{m}$. Also suppose that the events stated in Lemmas A.1 and A.5 holds. We can make these assumptions with high probability. For $j \in I_{\text{good}}$, we now know that $|a_j|^2 \leq (6/(\alpha m)) \|a\|_2^2 = O(1)$. Thus,

$$|b_j| \geq C|a_j| \cdot \min(1/|a_j|, 1) \geq \delta |a_j|$$

where $\delta = C \min(1, \sqrt{\alpha/6}/C')$. By the same argument as in the proof of Theorem A.6, except replacing $(c\sqrt{m})/\|a\|_2$ by $\delta$, we get that

$$\epsilon \|a\|_2^2 \leq \delta^{-1} \|a\|_2 \cdot \|A_U u\|_2.$$

Thus, $\|a\|_2 \leq \epsilon^{-1}\delta^{-1} \|A_U u\|_2$. By the isometry property of $A_U^T$ on its row space (Corollary G.5), we get

$$\|a\|_2 \leq \frac{\tau^2}{T\epsilon\delta\sqrt{m}} \|A_U^T A_U u\|_2 = \frac{c'}{\sqrt{m}} \|A_U^T b\|_2$$

for an appropriate constant $c'$. Since $a = A(\breve{x} - x^*)$ and $A_U$ is a $\sqrt{m}$-isometry up to constants (Theorem G.1), we get

$$\|\breve{x} - x^*\|_2 \leq \frac{\|a\|_2}{\tau} \leq \frac{c'}{\tau m} \|A_U^T b\|_2.$$

By Equation 4 and bounds on the other terms of Equation 4, the RHS of this inequality is $O(\lambda\sqrt{k} + \sqrt{(k \log n)/m})$.

$\square$

## J.7 Proof of Lemma B.1

For $1 \leq j \leq m$ we have by Lemma I.3 that

$$(\mathbb{E} R_{A_j,\breve{x}})^2 \leq \mathbb{E}[R_{A_j,\breve{x}}^2] \leq 2 \log \frac{1}{\gamma_S(A_j \breve{x})} + 4.$$

By Lemma I.2, we have

$$\log \frac{1}{\gamma_S(A_j \breve{x})} \leq 2 \log \frac{1}{\gamma_S(A_j x^*)} + |A_j \breve{x} - A_j x^*|^2 + 2.$$

Therefore summing over all $j \in [m]$,

$$\|\mathbb{E} R_{A,\breve{x}}\|_2^2 \leq 4 \sum_{j=1}^{m} \log \frac{1}{\gamma_S(A_j x^*)} + 2 \|A(\breve{x} - x^*)\|_2^2 + 8m.$$

Lemma I.4 bounds the first term. Theorems A.7 and G.1 bound the second: with high probability,

$$\|A(\breve{x} - x^*)\|_2 \leq 2T(\lambda \sqrt{km} + C'' \sqrt{k \log n}).$$

Thus,

$$\|\mathbb{E}[R_{A,\breve{x}}]\|_2^2 \leq 8m \log(1/\alpha) + 8m + 8T^2 \lambda^2 km + 8(TC'')^2 k \log n$$

with high probability. Under the assumptions $\lambda \leq c''_{\text{reg}}/\sqrt{k}$ and $m \geq M'' k \log n$, this quantity is $O(m)$.

## J.8  Proof of Lemma B.2

Draw $m$ samples from the distribution $R_{A_j, x^*}$ as follows: pick $a \sim N(0,1)^n$ and $\eta \sim N(0,1)$. Keep sample $\eta$ if $a^T x^* + \eta \in S$; otherwise reject. We want to bound $\eta_1^2 + \cdots + \eta_m^2$. Now consider the following revised process: keep all the samples, but stop only once $m$ samples satisfy $a^T x^* + \eta \in S$. Let $t$ be the (random) stopping point; then the random variable $\eta_1^2 + \cdots + \eta_t^2$ defined by the new process stochastically dominates the random variable $\eta_1 + \cdots + \eta_m^2$ defined by the original process.

But in the new process, each $\eta_i$ is Gaussian and independent of $\eta_1, \ldots, \eta_{i-1}$. With high probability, $t \leq 2m/\alpha$ by a Chernoff bound. And if $\eta'_1, \ldots, \eta'_{2m/\alpha} \sim N(0,1)$ are independent then

$$\eta'^2_1 + \cdots + \eta'^2_{2m/\alpha} \leq 4m/\alpha$$

with high probability, by concentration of norms of Gaussian vectors. Therefore $\eta_1^2 + \cdots + \eta_t^2 \leq 4m/\alpha$ with high probability as well.

## J.9  Proof of Theorem B.3

Set $\sigma = 4(\sqrt{c} + \sqrt{c_y})$, where $c$ and $c_y$ are the constants in Lemmas B.1 and B.2. Set $M = \max(16T^2 C''^2 (\sigma + 1)^2/\sigma^2, \sigma^2/c''^2_{\text{reg}})$. Note that $M$ is chosen sufficiently large that $\lambda = \sigma\sqrt{(\log n)/m} \leq c''_{\text{reg}}/\sqrt{k}$.

By Theorem A.7, we have with high probability that the following event holds, which we call $E_{\text{close}}$:

$$\|x^* - \breve{x}\|_2 \leq C''(\lambda \sqrt{k} + \sqrt{(k \log n)/m}) = C''(\sigma + 1)\sqrt{\frac{k \log n}{m}}.$$

Now notice that

$$\mathbb{E} Z_{A,\breve{x}} - y = A(\breve{x} - x^*) + \mathbb{E} R_{A,\breve{x}} - (y - Ax^*).$$

If $E_{\text{close}}$ holds, then by Theorem G.1, we get $\|A(\breve{x} - x^*)\|_2 \leq TC''(\sigma+1)\sqrt{k \log n}$. By Lemma B.1, with high probability $\|\mathbb{E} R_{A,\breve{x}}\|_2 \leq \sqrt{cm}$. And by Lemma B.2, with high probability $\|y - Ax^*\|_2 \leq \sqrt{c_y m}$. Therefore

$$\|\mathbb{E} Z_{A,\breve{x}} - y\|_2 \leq TC''(\sigma + 1)\sqrt{k \log n} + \sqrt{c_y m} + \sqrt{cm} \leq \frac{\sigma}{2}\sqrt{m}$$

where the last inequality is by choice of $M$ and $\sigma$. Thus, the event

$$E : \|\mathbb{E} Z_{A,\breve{x}} - y\|_2 \leq \frac{\sigma}{2}\sqrt{m}$$

occurs with high probability.

Suppose that event $E$ occurs. Now note that $A_{U^c}^T$ has independent Gaussian entries. Fix any $i \in U_c$; since $(A^T)_i$ is independent of $A_U$, $\breve{x}$, and $y$, the dot product

$$(A^T)_i(\mathbb{E}Z_{A,\breve{x}} - y)$$

is Gaussian with variance $\|\mathbb{E}Z_{A,\breve{x}} - y\|_2^2 \leq \sigma^2 m/4$. Hence, $\breve{z}_i = \frac{1}{\lambda m}(A^T)_i(\mathbb{E}Z_{A,\breve{x}} - y)$ is Gaussian with variance at most $(\sigma^2 m/4)/(\lambda m)^2 = 1/(4 \log n)$. So

$$\Pr[|\breve{z}_i| \geq 1] \leq 2e^{-2 \log n} \leq \frac{2}{n^2}.$$

By a union bound,

$$\Pr[\|\breve{z}_{U^c}\|_\infty \geq 1] \leq \frac{2}{n}.$$

So the event $\|\breve{z}_{U^c}\| < 1$ holds with high probability.

## J.10  Proof of Theorem B.4

We know from Theorem B.3 that $\left\|\frac{1}{m}A_{U^c}^T(\mathbb{E}[Z_{A,\breve{x}}] - y)\right\|_\infty \leq \lambda/3$. So it suffices to show that

$$\frac{1}{m}\left\|A_{U^c}^T(\mathbb{E}[Z_{A,X}] - y) - A_{U^c}^T(\mathbb{E}[Z_{A,\breve{x}}] - y)\right\|_\infty \leq \frac{\lambda}{6}.$$

Thus, we need to show that

$$\frac{1}{m}|(A^T)_i(\mathbb{E}[Z_{A,X}] - \mathbb{E}[Z_{A,\breve{x}}])| \leq \frac{\lambda}{6}$$

for all $i \in U^c$. Fix one such $i$. Then by Lemma A.2,

$$\begin{aligned}
\|\mathbb{E}[Z_{A,X}] - \mathbb{E}[Z_{A,\breve{x}}]\|_2^2 &= \sum_{i=1}^m (\mu_{A_i X} - \mu_{A_i \breve{x}})^2 \\
&= \sum_{i=1}^m \left(\int_{A_i X}^{A_i \breve{x}} \mathrm{Var}(Z_t)\, dt\right)^2 \\
&\leq \sum_{i=1}^m (A_i X - A_i \breve{x})^2 \cdot \sup_{t \in [A_i X, A_i \breve{x}]} \mathrm{Var}(Z_t).
\end{aligned}$$

By Lemma I.4, we have

$$\sum_{j=1}^m \log \frac{1}{\gamma_S(A_j x^*)} \leq 2m \log(1/\alpha)$$

with high probability over $A$. Assume that this inequality holds, and assume that $\|X - \breve{x}\|_2 \leq 1$ and $\|\breve{x} - x^*\|_2 \leq 1$, so that $\|X - x^*\|_2 \leq 2$. Then by Theorem G.1, $\|A(X - x^*)\|_2 \leq 2T\sqrt{m}$. By Lemma I.2, for every $j \in [m]$ and every $t \in [A_j X, A_j \breve{x}]$,

$$\log \frac{1}{\gamma_S(t)} \leq 2 \log \frac{1}{\gamma_S(A_j x^*)} + |t - A_j x^*|^2 + 2 \leq cm$$

for a constant $c$. Hence, by Lemma I.3,

$$\mathrm{Var}(Z_t) \leq \mathbb{E}[(Z_t - t)^2] \leq 2 \log \frac{1}{\gamma_S(t)} + 4 \leq 2cm + 4.$$

We conclude that

$$\|\mathbb{E}[Z_{A,X}] - \mathbb{E}[Z_{A,\breve{x}}]\|_2^2 \leq (2cm + 4)\|AX - A\breve{x}\|_2^2 \leq (2cm + 4)\frac{T^2 m}{m^2} \leq O(1).$$

Additionally, $\left\|(A^T)_i\right\|_2 \leq T\sqrt{m}$ with high probability. Thus, Cauchy-Schwarz entails that

$$\frac{1}{m}|(A^T)_i(\mathbb{E}[Z_{A,X}] - \mathbb{E}[Z_{A,\breve{x}}])| \leq \frac{1}{m}T\sqrt{m} \cdot O(1) \leq \frac{\lambda}{6}$$

for large $n$.

### J.11 Proof of Lemma 5.4

Note that
$$\|A\breve{x} - y\|_2 \leq \|A(\breve{x} - x^*)\|_2 + \|Ax^* - y\|_2.$$
With high probability, $\|\breve{x} - x^*\|_2 \leq 1$. Theorem G.1 gives that $\|A(\breve{x} - x^*)\|_2 \leq T\sqrt{m}$. Furthermore, $\|Ax^* - y\|_2 \leq 2\sqrt{m/\alpha}$ by Lemma B.2.

### J.12 Proof of Lemma 5.5

With high probability $\breve{x} \in \mathscr{E}_r$ by the above lemma. Note that $\|x^{(0)}\|_1 \leq \|\breve{x}\|_1$, and $\|A(x^{(0)} - \breve{x})\|_2 \leq 2r\sqrt{m}$. Set $\rho = \min(\tau/(4T), 1/3)$ and $s = k(1 + 1/\rho^2)$. If $m \geq Mk\log n$ for a sufficiently large constant $M$, then by Corollary G.6, $A/\sqrt{m}$ with high probability satisfies the $s$-Restricted Isometry Property. Then by Theorem D.1 (due to [6], but reproduced here for completeness), it follows that $\|x^{(0)} - \breve{x}\|_2 \leq O(1)$.

### J.13 Proof of Lemma 5.6

Note that $\mathrm{sign}(x^{(t)})$ is a subgradient for $\|x\|_1$ at $x = x^{(t)}$. Furthermore, for fixed $A$,
$$\mathbb{E}\left[A_j(z^{(t)} - y_j)\Big|x^{(t)}\right] = \frac{1}{m}\sum_{j'=1}^m A_{j'}(\mathbb{E}Z_{A_{j'},x^{(t)}} - y_{j'}) = \nabla\,\mathrm{nll}(x^{(t)}; A, y).$$

It follows that
$$\mathbb{E}[v^{(t)}|x^{(t)}] = \mathbb{E}\left[A_j(z^{(t)} - y_j)\Big|x^{(t)}\right] + \mathrm{sign}(x^{(t)})$$
is a subgradient for $f(x)$ at $x = x^{(t)}$.

We proceed to bounding $\mathbb{E}[\|v^{(t)}\|_2^2|x^{(t)}]$. By definition of $v^{(t)}$,
$$\left\|v^{(t)}\right\|_2^2 \leq 2\left\|A_j(z^{(t)} - y_j)\right\|_2^2 + 2\left\|\lambda \cdot \mathrm{sign}(x^{(t)})\right\|_2^2$$
where $j \in [m]$ is uniformly random, and $z^{(t)}|x^{(t)} \sim Z_{A_j,x^{(t)}}$. Since $\left\|\lambda \cdot \mathrm{sign}(x^{(t)})\right\|_2^2 = o(n)$ it remains to bound the other term. We have that
$$\mathbb{E}[\left\|A_j(z^{(t)} - y_j)\right\|_2^2|x^{(t)}] = \frac{1}{m}\sum_{j'=1}^m \mathbb{E}[\left\|A_{j'}(Z_{A_{j'},x^{(t)}} - y_{j'})\right\|_2^2].$$

With high probability, $\|A_i\|_2^2 \leq 2n$ for all $i \in [m]$. Thus,
$$\mathbb{E}[\left\|A_j(z^{(t)} - y_j)\right\|_2^2|x^{(t)}] \leq \frac{n}{m}\sum_{j'=1}^m \mathbb{E}[(Z_{A_{j'},x^{(t)}} - y_{j'})^2].$$

Now
$$\sum_{i=1}^m \mathbb{E}[(Z_{A_i,x^{(t)}} - y_i)^2] \leq 2\sum_{i=1}^m (A_i x^{(t)} - y_i)^2 + 2\sum_{i=1}^m \mathbb{E}[(A_i x^{(t)} - Z_{A_i,x^{(t)}})^2].$$
The first term is bounded by $2r^2m$ since $x^{(t)} \in \mathscr{E}_r$. Additionally,
$$\left\|A(x^{(t)} - x^*)\right\|_2 \leq \left\|Ax^{(t)} - y\right\|_2 + \|Ax^* - y\|_2 \leq 2r\sqrt{m}$$
since $x^{(t)}, x^* \in \mathscr{E}_r$. Therefore the second term is bounded as
$$2\sum_{i=1}^m \mathbb{E}[R_{A_i,x^{(t)}}^2] \leq 4\sum_{i=1}^m \log\left(\frac{1}{\gamma_S(A_i x^{(t)})}\right) + 8m$$
$$\leq 8\sum_{i=1}^m \log\left(\frac{1}{\gamma_S(A_i x^*)}\right) + 4\left\|A(x^{(t)} - x^*)\right\|_2^2 + 16m$$
$$\leq 64\log(1/\alpha)m + 16r^2m + 80m.$$

where the first and second inequalities are by Lemmas I.3 and Lemma I.2, and the third inequality is by Lemma I.4. Putting together the two bounds, we get

$$\sum_{i=1}^{m} \mathbb{E}[(Z_{A_i,x^{(t)}} - y_i)^2] \leq O(m),$$

from which we conclude that $\mathbb{E}[\|v^{(t)}\|_2^2 \,|\, x^{(t)}] \leq O(n)$. The law of total expectation implies that $\mathbb{E}[\|v^{(t)}\|_2^2] \leq O(n)$ as well.

## J.14   Proof of Lemma 5.7

We need to show that $f_{\mathbb{R}^U}$ is $\zeta$-strongly convex near $\breve{x}$. Since $\|x\|_1$ is convex, it suffices to show that $\mathrm{nll}(x; A, y)_{\mathbb{R}^U}$ is $\zeta$-strongly convex near $\breve{x}$. The Hessian of $\mathrm{nll}(x; A, y)|_{\mathbb{R}^U}$ is

$$H_U(x; A, y) = \frac{1}{m} \sum_{j=1}^{m} A_{j,U}^T A_{j,U} \operatorname{Var}(Z_{A_j,x}).$$

Hence, it suffices to show that

$$\frac{1}{m} \sum_{j=1}^{m} A_{j,U}^T A_{j,U} \operatorname{Var}(Z_{A_j,x}) \succeq \zeta I$$

for all $x \in \mathbb{R}^n$ with $\mathrm{supp}(x) \subseteq U$ and $\|x - \breve{x}\|_2 \leq 1$. Call this region $\mathscr{B}$. With high probability over $A$ we can deduce the following.

**(i)** By Theorem A.7, we have $\|\breve{x} - x^*\|_2 \leq d\sqrt{(k \log n)/m}$. As $\|x - \breve{x}\|_2 \leq 1$ for all $x \in \mathscr{B}$, we get $\|A(\breve{x} - x)\|_2^2 \leq T^2(d+1)^2 m$ for all $x \in \mathscr{B}$.

**(ii)** By the proof of Lemma A.4, the number of $j \in [m]$ such that $\gamma_S(A_j x^*) \leq \alpha/2$ is at most $(1 - \alpha/3)m$.

Fix $x \in \mathscr{B}$, and define $J_x \subseteq [m]$ to be the set of indices

$$J_x = \{j \in [m] : \gamma_S(A_j x^*) \geq \alpha/2 \wedge |A_j(x - x^*)|^2 \leq (6/\alpha)T^2(d+1)^2.\}$$

For any $j \in J_x$,

$$\log \frac{1}{\gamma_S(A_j x)} \leq 2 \log \frac{1}{\gamma_S(A_j x^*)} + |A_j(x - x^*)|^2 + 2 \leq \log(2/\alpha) + (6/\alpha)T^2(d+1)^2 + 2.$$

Thus,

$$\operatorname{Var}(Z_{A_j,x}) \geq C\gamma_S(A_j x)^2 \geq e^{-\log(2/\alpha) - (6/\alpha)T^2(d+1)^2 - 2} = \Omega(1).$$

Let $\delta$ denote this lower bound—a positive constant. By **(i)** and **(ii)**, $|J_x| \geq (\alpha/6)m$, so by Theorem G.1,

$$H_U(x; A, y) = \frac{1}{m} \sum_{j=1}^{m} A_{j,U}^T A_{j,U} \operatorname{Var}(Z_{A_j,x}) \succeq \frac{\delta}{m} A_{J_x,U}^T A_{J_x,U} \succeq \delta\tau I$$

as desired.

## J.15   Proof of Lemma 5.8

Let $t = \|(x - \breve{x})_U\|_2$. Define $w = \breve{x} + (x - \breve{x})\min(t^{-1}/m, 1)$. Also define $w' = [w_U; 0_{U^c}] \in \mathbb{R}^n$. Then $\|(w - \breve{x})_U\|_2 \leq 1/m$, so

$$\|(\nabla \mathrm{nll}(w'; A, y))_{U^c}\|_\infty \leq \frac{\lambda}{2}.$$

Therefore $w_i \cdot (\nabla \mathrm{nll}(w'; A, y))_i \leq (\lambda/2)|w_i|$ for all $i \in U^c$, so

$$\begin{aligned}
f(w) - f(w') &= (\mathrm{nll}(w; A, y) - \mathrm{nll}(w'; A, y)) + \lambda(\|w\|_1 - \|w'\|_1) \\
&\geq (w - w') \cdot \nabla \mathrm{nll}(w'; A, y) + \lambda \|w_{U^c}\|_1 \\
&\geq \frac{\lambda}{2} \|w_{U^c}\|_1.
\end{aligned}$$

Additionally, since $\|w' - \check{x}\|_2 \le 1$ and $\mathrm{supp}(w') \subseteq U$, we know that

$$f(w') - f(\check{x}) \ge \frac{\zeta}{2} \|w' - \check{x}\|_2^2.$$

Adding the second inequality to the square of the first inequality, and lower bounding the $\ell_1$ norm by $\ell_2$ norm,

$$\frac{1}{2}(f(w) - f(\check{x}))^2 + \frac{1}{2}(f(w) - f(\check{x})) \ge \frac{1}{2}(f(w) - f(w'))^2 + \frac{1}{2}(f(w') - f(\check{x}))$$

$$\ge \frac{\lambda^2}{8} \|w_{U^c}\|_2^2 + \frac{\zeta}{4} \|w' - \check{x}\|_2^2$$

$$\ge \frac{\lambda^2}{8} \|(w - \check{x})_{U^c}\|_2^2 + \frac{\zeta}{4} \|(w - \check{x})_U\|_2^2$$

$$\ge \min\left(\frac{\lambda^2}{8}, \frac{\zeta}{4}\right) \|w - \check{x}\|_2^2$$

Since $f(x) - f(\check{x}) \le 1$, by convexity $f(w) - f(\check{x}) \le 1$ as well. Hence,

$$f(w) - f(\check{x}) \ge \frac{1}{2}(f(w) - f(\check{x}))^2 + \frac{1}{2}(f(w) - f(\check{x})) \ge \min\left(\frac{\lambda^2}{8}, \frac{\zeta}{4}\right) \|w - \check{x}\|_2^2. \qquad (7)$$

We distinguish two cases:

1. If $t \le 1/m$, then $w = x$, and it follows from Equation 7 that

$$f(x) - f(\check{x}) \ge \min\left(\frac{\lambda^2}{8}, \frac{\zeta}{4}\right) \|x - \check{x}\|_2^2$$

   as desired.

2. If $t \ge 1/m$, then $\|(w - \check{x})_U\|_2 = 1/m$, and thus $\|w - \check{x}\|_2 \ge 1/m$. By convexity and this bound,

$$f(x) - f(\check{x}) \ge f(w) - f(\check{x}) \ge \min\left(\frac{\lambda^2}{8}, \frac{\zeta}{4}\right) \frac{1}{m^2},$$

   which contradicts the lemma's assumption for a sufficiently small constant $c_f > 0$.

### J.16 Proof of Theorem 5.9

By Lemmas 5.4, 5.5, and 5.6, we are guaranteed that $\check{x} \in \mathscr{E}_r$, $\|x^{(0)} - \check{x}\|_2^2 \le O(1)$, and $\mathbb{E}[\|v^{(t)}\|_2^2] \le O(n)$ for all $t$. Thus, applying Theorem H.1 with projection set $\mathscr{E}_r$, step count $T = m^6 n^2$, and step size $\eta = 1/\sqrt{Tn}$ gives $\mathbb{E}[f(\bar{x})] - f(\check{x}) \le O(1/(m^3 n))$. Since $f(\bar{x}) - f(\check{x})$ is nonnegative, Markov's inequality gives

$$\Pr[f(\bar{x}) - f(\check{x}) \le c_f (\log n)/m^3] \ge 1 - 1/n.$$

From Theorem 5.8 we conclude that $\|\bar{x} - \check{x}\|_2 \le O(1/m)$ with high probability.

## K Efficient sampling for union of intervals

In this section, in Lemma K.4, we see that when $S = \cup_{i=1}^r [a_i, b_i]$, with $a_i, b_i \in \mathbb{R}$, then Assumption II holds with $T(\gamma_S(t)) = \mathrm{poly}(\log(1/\gamma_S(t)), r)$. The only difference is that instead of exact sampling we have approximate sampling, but the approximation error is exponentially small in total variation distance and hence it cannot affect any algorithm that runs in polynomial time.

**Definition K.1** (EVALUATION ORACLE). Let $f : \mathbb{R} \to \mathbb{R}$ be an arbitrary real function. We define the *evaluation oracle* $\mathscr{E}_f$ of $f$ as an oracle that given a number $x \in \mathbb{R}$ and a target accuracy $\eta$ computes an $\eta$-approximate value of $f(x)$, that is $|\mathscr{E}_f(x) - f(x)| \le \eta$.

**Lemma K.2.** *Let $f : \mathbb{R} \to \mathbb{R}_+$ be a real increasing and differentiable function and $\mathscr{E}_f(x)$ an evaluation oracle of $f$. Let $\ell \le f'(x) \le L$ for some $\ell, L \in \mathbb{R}_+$. Then we can construct an algorithm that implements the evaluation oracle of $f^{-1}$, i.e. $\mathscr{E}_{f^{-1}}$. This implementation on input $y \in \mathbb{R}_+$ and input accuracy $\eta$ runs in time $T$ and uses at most $T$ calls to the evaluation oracle $\mathscr{E}_f$ with inputs $x$ with representation length $T$ and input accuracy $\eta' = \eta/\ell$, with $T = \mathrm{poly} \log(\max\{|f(0)/y|, |y/f(0)|\}, L, 1/\ell, 1/\eta)$.*

*Proof of Lemma K.2.* Given a value $y \in \mathbb{R}_+$ our goal is to find an $x \in \mathbb{R}$ such that $f(x) = y$. Using doubling we can find two numbers $a, b$ such that $f(a) \leq y - \eta'$ and $f(b) \geq y + \eta'$ for some $\eta'$ to be determined later. Because of the lower bound $\ell$ on the derivative of $f$ we have that this step will take $\log((1/\ell) \cdot \max\{|f(0)/y|, |y/f(0)|\})$ steps. Then we can use binary search in the interval $[a, b]$ where in each step we make a call to the oracle $\mathcal{E}_f$ with accuracy $\eta'$ and we can find a point $\hat{x}$ such that $|f(x) - f(\hat{x})| \leq \eta'$. Because of the upper bound on the derivative of $f$ we have that $f$ is $L$-Lipschitz and hence this binary search will need $\log(L/\eta')$ time and oracle calls. Now using the mean value theorem we get that for some $\xi \in [a, b]$ it holds that $|f(x) - f(\hat{x})| = |f'(\xi)| |x - \hat{x}|$ which implies that $|x - \hat{x}| \leq \eta'/\ell$, so if we set $\eta' = \ell \cdot \eta$, the lemma follows. $\qquad\square$

Using the Lemma K.2 and the Proposition 3 of [7] it is easy to prove the following lemma.

**Lemma K.3.** *Let $[a, b]$ be a closed interval and $\mu \in \mathbb{R}$ such that $\gamma_{[a,b]}(\mu) = \alpha$. Then there exists an algorithm that runs in time* poly $\log(1/\alpha, \zeta)$ *and returns a sample of a distribution $\mathscr{D}$, such that $d_{\mathrm{TV}}(\mathscr{D}, N(\mu, 1; [a, b])) \leq \zeta$.*

*Proof Sketch.* The sampling algorithm follows the steps: (1) from the cumulative distribution function $F$ of the distribution $N(\mu, 1; [a, b])$ define a map from $[a, b]$ to $[0, 1]$, (2) sample uniformly a number $y$ in $[0, 1]$ (3) using an evaluation oracle for the error function, as per Proposition 3 in [7], and using Lemma K.2 compute with exponential accuracy the value $F^{-1}(y)$ and return this as the desired sample. $\qquad\square$

Finally using again Proposition 3 in [7] and Lemma K.3 we can get the following lemma.

**Lemma K.4.** *Let $[a_1, b_1]$, $[a_2, b_2]$, ..., $[a_r, b_r]$ be closed intervals and $\mu \in \mathbb{R}$ such that $\gamma_{\cup_{i=1}^r [a_i, b_i]}(\mu) = \alpha$. Then there exists an algorithm that runs in time* poly$(\log(1/\alpha, \zeta), r)$ *and returns a sample of a distribution $\mathscr{D}$, such that $d_{\mathrm{TV}}(\mathscr{D}, N(\mu, 1; \cup_{i=1}^r [a_i, b_i])) \leq \zeta$.*

*Proof Sketch.* Using Proposition 3 in [7] we can compute $\hat{\alpha}_i$ which estimated with exponential accuracy the mass $\alpha_i = \gamma_{[a_i, b_i]}(\mu)$ of every interval $[a_i, b_i]$. If $\hat{\alpha}_i/\alpha \leq \zeta/3r$ then do not consider interval $i$ in the next step. From the remaining intervals we can choose one proportionally to $\hat{\alpha}_i$. Because of the high accuracy in the computation of $\hat{\alpha}_i$ this is $\zeta/3$ close in total variation distance to choosing an interval proportionally to $\alpha_i$. Finally after choosing an interval $i$ we can run the algorithm of Lemma K.3 with accuracy $\zeta/3$ and hence the overall total variation distance from $N(\mu, 1; \cup_{i=1}^r [a_i, b_i])$ is at most $\zeta$. $\qquad\square$