[Reviews · NeurIPS 2020]

Review 1

Summary and Contributions: This work studies high-dimensional linear regression under a constraint imposed by a truncation set S, where any pair (a, y) is hidden if y is not in S. A computationally efficient algorithm is proposed to recover the groundtruth signal, and a near-optimal statistical error rate is established.

Strengths: - The paper is clearly written, and connection to prior works is elaborated in detail. Thus it is easy to follow the main contribution of the work. - Both computational and statistical results are presented. - Authors also make significant efforts to explain the primary proof idea (which is appreciated).

Weaknesses: - My major concern is why the problem is difficult. Assumption 1 literally enforces that the adversary cannot pick arbitrary S, but only those such that a constant alpha-fraction of the observations are hidden/removed. Thus, suppose before removal we have a total of m samples (a, y). After removal it reduces to alpha * m pairs (a, y), which still suffices for accurate recovery provided that m = O(k log n). - It is not convincing to me that the sample complexity in Theorem 3.1 is near-optimal. I know that O(k log n) is near-optimal, but does your result really imply such bound? Observe that your success probability is 1 - 1/log n, which has a very slow convergence to 1. In contrast, almost all existing works provide a strong probability of 1 - n^{-100}, or even 1 - exp(-n). If you rephrase Theorem 3.1 in a way that with probability 1- delta, || x_bar - x^* || < f(m, k, n, delta) What is the expression of the function f? - In the rest of the paper, the success probability is superceded with "high probability", which I do not believe comforms common practice. Again, note that 1-1/log n does not suffice for a high probability argument since it carries out serious issue on sample complexity. - I am disappointed that the proof of Theorem 3.1 is nowhere found, and the organization of the supplement is quite a mess. I was not able to track the true sample complexity under a real high probability event. - The term "adversarial noise" is misleading. You are not handling arbitrary noise, e.g. you cannot slightly tailor your algorithm to gross outliers. - PDW in Wainwright 2009 requires a lower bound on |xhat - x|, but this paper asks for an upper bound. Is there any intuition behind the essential difference? - I find Lemmas 5.7 and 5.8 are hard to follow. How can you ensure/test for any x, f(x) - f(xhat) < log n / m^3 ? If the purpose of Section 5.2 is to provide faster rate of SGD, why don't you show that the Hessian matrix (Lemma 5.1) has a lower bound on its smallest singular value when restricted on sparse directions? My overall feeling is that the paper presents a dense set of results to extend a previous work on truncated linear regression to the high-dimensional regime. Yet, most of the key techniques, e.g. log-likelihood of truncated linear regression, projected SGD, have been established in Daskalakis 2019. The main difference lies in an application of primal-dual witness, but was also known due to Wainwright 2009. The discussion in the main body, and the proof in the supplement, are hard to follow. I believe many of the results have been proved in Daskalakis 2019. So references to lemmas thereof (maybe with a short paragraph explaining the difference) should serve the purpose. ---updates after rebuttal--- Thank you for addressing my concerns. I suggest incorporating them into your revision. I still feel Section 5 is dense and hard to follow. The notation and terminology can also be improved. Overall this is a good submission, but it turns out the computational efficiency heavily relies on Assumption II.

Correctness: Not all. See Weakness.

Clarity: Yes.

Relation to Prior Work: Yes.

Reproducibility: Yes

Additional Feedback:


Review 2

Summary and Contributions: This paper looks at the problem of truncated linear regression where we see the labels depending on if they lie in some fixed known set for the case when the underlying true model is k sparse. They give an algorithm which gets optimal error guarantees for this setting as function of number of samples under standard assumptions. The algorithm is also computationally efficient with certain conditions on the truncation set.

Strengths: This paper looks at the problem of truncated linear regression where we see the labels depending on if they lie in some fixed known set for the case when the underlying true model is k sparse. They give an algorithm which gets optimal error guarantees with l2 squared error scaling linearly in the sparsity dimension for this setting as function of number of samples under standard assumptions. The algorithm is also computationally efficient with certain conditions on the truncation set. I think the problem of biased data is an important problem. The paper presents interesting results and seems like has strong technical contributions. The paper is also very well written clearly explaining the challenges in extending previous techniques and how they deal with them.

Weaknesses: Can the authors comment on how easy is this algorithm to implement in practice? What is the precise dependence of n and m in running time? -------- I have read the authors' response and would like to thank the authors for clarifying the runtime dependence.

Correctness: The proofs look correct at a first glance.

Clarity: Yes, the paper is very well written.

Relation to Prior Work: Yes, the paper has sufficiently discussed comparison with previous work.

Reproducibility: Yes

Additional Feedback: Yes, the paper has sufficiently discussed comparison with previous work.


Review 3

Summary and Contributions: The author propose to solve truncated sparse linear regression problem which has recently gained attention due to works by Daskalakis and Wainwright. However, their techniques do not directly apply to the current formulations and the authors propose new tools to solve the resulting problems such as lack of strong convexity, finding good initialization for the SGD approach etc.

Strengths: (A) Successfully solve the very relevant truncated linear regression problem and provide theoretical guarantees of l2 error of O(\sqrt(k log n)/m)). (B) Highly relevant for real-world settings where the data could be censored or truncated due to privacy reasons.

Weaknesses: (A) I did not see any experimental work which would have strengthened the work.

Correctness: Based on the reading of the main paper it seems right but did not check the proofs in the supplementary.

Clarity: The paper is clearly organized with prior work, issues in applying them to current problem and how they are solved with proposed approach.

Relation to Prior Work: Prior works are clearly discussed and built on.

Reproducibility: Yes

Additional Feedback: Update after discussion: Keeping the score since the contribution seems interesting and did not uncover any surprises in discussion phase.

[Author Response · NeurIPS 2020]

*We thank all reviewers for their comments. Minor points will be dealt with in the revised version of our article.*

**R#2: My major concern...provided that m = O(k log n).** The problem is actually much more subtle. It is not hard to show that the sparse vector is, in fact, *not identifiable even with infinitely many samples*, if the only assumption one makes is that a constant fraction of the measurements survive truncation. Here's a simple example: consider sampling (a) a Gaussian N(0,1), versus (b) a Gaussian N(0,2). We can define a randomized truncation procedure for situation (a), and for situation (b), such that the truncated samples from (a) are identically distributed to those from (b). Only a constant fraction of the observations are removed. Nonetheless, no estimation of the variance is possible. This can be recast as a linear regression problem with n=k=1; even for m arbitrarily large, recovery is impossible.

Another point we want to make here is why a two-step argument of the following form fails: (1) first show that, if a constant fraction of the measurements survive, then the measurement matrix corresponding to the surviving samples is incoherent (RIP/REC/whatever); (2) then use existing results in the literature implying that such incoherence conditions are sufficient for sparse recovery. This is, indeed, a natural approach but there is a subtle reason why it fails. In order to black box existing recovery results we need that the noise added to the measurements is independent of the measurement matrix. Unfortunately, in our case the measurement matrix of the surviving measurements depends on the noise, breaking the independence assumption, and thus invalidating this simple approach. It is true, however, that without measurement noise this approach works as we point out in Theorem 3.4.

**It is not convincing...expression of the function $f$?** We did not think about trying to improve the rate, since there are generic methods to amplify to $1/\mathrm{poly}(n)$ by repeatedly running the algorithm on independent samples, at the cost of $O(\log n)$ extra multiplicative factor in sample complexity. However, you are correct that this is suboptimal. The suboptimality is entirely due to a single bound (Lemma A.1), and it turns out that it is fairly straightforward to modify Lemma A.1 to achieve $1/\mathrm{poly}(n)$ failure rate, which leads to $1/\mathrm{poly}(n)$ failure rate in Theorem 3.1, matching the exact type of bounds known for the untruncated case. Specifically, Lemma A.1 relies on a first moment bound and Markov's inequality. We can instead prove sub-Gaussianity of the truncated Gaussian $y - \mathbb{E}[y]$ conditioned on $A$. We can then bound the sub-Gaussian variance proxy of $(A^{\mathrm{T}})_i \cdot (y - \mathbb{E}[y])$ (holding $A$ fixed) with high probability over $A$. Finally apply sub-Gaussian tail bounds, and sum over $i \in U$. With this modification, the algorithm in our paper and all statements that use the phrase "with high probability" achieve success probability $1 - n^{-100}$ with $O(k \log n)$ samples.

**I am disappointed that the proof...high probability event.** Theorem 3.1 follows immediately from the concatenation of Propositions 3.2 and 3.3, which we provide proofs for. We will include a statement to this effect in our next revision. Again, with the above modification the probability of success is $1 - 1/\mathrm{poly}(n)$ when $m = O(k \log n)$.

**The term "adversarial noise" is misleading...** We could say "arbitrary bounded noise."

**PDW in Wainwright 2009 requires a lower bound...difference?** We are not sure where Wainwright requires a lower bound on $|\hat{x} - x|$. Our paper proves an upper bound on the reconstruction error as a function of $n, k, m$.

**I find Lemmas 5.7 and 5.8...sparse directions?** Due to lack of (unrestricted) strong convexity, SGD only guarantees that $f(x^t)$, where $x^t$ is the $t$-th iterate, converges to $f(\hat{x})$ ($\hat{x}$ is optimum of regularized NLL program). Section 5.2 shows that in fact $x^t$ converges to $\hat{x}$. Specifically, Lemma 5.8 shows that if $f(x)$ is close to $f(\hat{x})$, then $x$ is close to $\hat{x}$. The algorithm never needs to test that $f(x) - f(\hat{x}) < \log n / m^3$. This condition is guaranteed by SGD. We do lower bound the Hessian's smallest singular value in the sparse support of $x^*$ (this is Lemma 5.7). However, this is not sufficient to prove that SGD converges, since SGD's iterates are dense.

**My overall feeling...Wainwright 2009** We are very familiar with the results of both [Daskalakis et al. 2019] and [Wainwright 2009]. While we use primal-dual witness and log-likelihood, there are a number of innovations required for high-dimensional truncated regression but not in the prior work, as explained in Section 1.1 (overview of proofs and techniques). On a technical level, we cite three lemmas from [Daskalakis et al. 2019]. Aside from these, the results of the prior work do not have useful instantiations in the high-dimensional setting. The structure of our proof is quite distinct from that of [Daskalakis et al. 2019], since in their case the special projection set yields all the properties needed for SGD convergence, but in our case there can be no such projection set.

**R# 3: Can the authors comment...running time?** The algorithm is fairly straightforward to implement; each update of the SGD requires a sampling step and a projection. In comparison to [Daskalakis et al. 2019], the sampling step is trickier (described in Section K of the supplementary); however the projection is simpler. From our theoretical results we achieve polynomial dependence of $\tilde{O}(nk^6)$ for $m = O(k \log n)$. We expect that in practice fewer iterations are needed, but we do not have experimental results to confirm or deny this belief.

**R# 4:** (regarding experimental work) After this work in which we established the necessary theoretical framework for this problem, we agree that the next relevant step is to try our method both on simulated and on real-world data where truncation occurs.

[Meta-Review · NeurIPS 2020]

The truncated setting is interesting both in practice and theoretically. The reviewers uniformly felt that this is an interesting paper, and a good contribution to the community.